# Unlocking the functional potential of polyploid yeasts

Simone Mozzachiodi [1,2✉], Kristoffer Krogerus [3], Brian Gibson[3,5], Alain Nicolas[1,2,4] & Gianni Liti [1✉]

Breeding and domestication have generated widely exploited crops, animals and microbes. However, many *Saccharomyces cerevisiae* industrial strains have complex polyploid genomes and are sterile, preventing genetic improvement strategies based on breeding. Here, we present a strain improvement approach based on the budding yeasts' property to promote genetic recombination when meiosis is interrupted and cells return-to-mitotic-growth (RTG). We demonstrate that two unrelated sterile industrial strains with complex triploid and tetraploid genomes are RTG-competent and develop a visual screening for easy and high-throughput identification of recombined RTG clones based on colony phenotypes. Sequencing of the evolved clones reveal unprecedented levels of RTG-induced genome-wide recombination. We generate and extensively phenotype a RTG library and identify clones with superior biotechnological traits. Thus, we propose the RTG-framework as a fully non-GMO workflow to rapidly improve industrial yeasts that can be easily brought to the market.

[1] Université Côte d'Azur, CNRS, INSERM, IRCAN, Nice, France. [2] Meiogenix, Paris 75011, France. [3] VTT Technical Research Centre of Finland Ltd, Espoo, Finland. [4] Institut Curie, Centre de Recherche, CNRS-UMR3244, PSL Research University, Paris 75005, France. [5]Present address: Institute for Food Technology and Food Chemistry Department of Brewing and Beverage Technology, Technical University, 13353 Berlin, Germany. ✉email: simone.mozzachiodi@univ-cotedazur.fr; gianni.liti@unice.fr

Humans have selected and improved organisms for centuries, leaving profound hallmarks of domestication in their genomes and lifestyles[1]. Selective breeding, in which hybrids with improved performance are generated, was one of humanity's first biotechnology advances and it remains widely applied[2]. The advent of new genetic engineering techniques enabled to directly manipulate core biological traits relevant for human activities[3]. However, the exploitation of genetically modified organisms (GMOs) in the food sector remains controversial, highly regulated and restricted in many countries.

Humans unwittingly domesticated yeasts of the genus *Saccharomyces* since the earliest food and beverages fermentations practices[4–6]. Since then, novel fermentation processes selected new yeast strains that are still used nowadays[7]. Nevertheless, there is a pressing interest across different sectors that exploit *Saccharomyces* yeasts to create new strain variants able to better tolerate stresses encountered during the industrial fermentations or increase the yield of desired compounds. Newly designed de novo lab-hybrids that combine different *Saccharomyces* species have displayed good or superior fermenting properties compared to common commercial strains. Recently, other methods based on breeding have further expanded the capability of generating genetic diversity in *Saccharomyces* species in diploid interspecies hybrids by passing through intermediate fertile tetraploid[8] or by generating hybrids with higher ploidy through iterative hybrid production (iHyPr)[9]. This suggests that indeed commercial strains are not yet optimized, leaving room for improvement despite the long domestication period[10–12]. However, attempts to cross industrial yeasts through selective breeding or induce genomic recombination through meiosis are often unsuccessful due to the inherent sterility of the strains, which is a hallmark of yeast domestication[6,13] and breeding them relies on a CRISPR/Cas9 engineering[14]. The domestication syndrome might have derived from the lack of selection on sexual reproduction, random genetic drift or adaptation to specific fermentation niches, leading to the accumulation of punctuated deleterious loss-of-functions (LOF) alleles that inactivate genes involved in the gametogenesis (sporulation in yeast). In addition, the genomes of industrial strains often have features such as extreme sequence divergence between the subgenomes (namely heterozygosity), structural rearrangements, aneuploidy and polyploidy all of which are known to contribute to sterility[15]. Classic examples are strains used in beer production or in the baking industry sharing different ancestries that have repeatedly converged toward polyploid complex genomes[5,16,17]. However, tetraploid strains derived from designed crosses perform a correct chromosome segregation and produce viable gametes[18] in contrast to triploid strains. Therefore, the genetic basis driving this extreme sterility are not yet fully understood and multiple factors likely contribute to impair the sexual reproduction of industrial polyploid strains. Thus, other approaches to improve sterile industrial strains have been proposed[19] because directly fixing the lack of a complete sexual reproduction remains challenging.

Recently, we demonstrated that aborting meiosis in laboratory-derived sterile diploid hybrids and returning them to mitotic growth, a process called return-to-growth (RTG), promoted evolution by enabling recombination between subgenomes resulting in phenotypic variability in the evolved samples[20,21]. RTG is induced when cells that have entered meiosis but are not yet committed to complete it are shifted back to a nutrient-rich environment, although some mutants can maintain an extended uncommited state[22,23]. As in normal meiosis, after DNA replication Spo11p induces multiple genome-wide double-stranded breaks (DSBs) that lead to the formation of intermediate recombinant molecules[24]. These molecules are then resolved upon the re-entering of the mitotic cell cycle, resulting in

dispersed LOH tracts that derive from the segregation of the chromatids in the mother and daughter cells. Despite generating multiple dispersed LOH tracts, the RTG process is not mutagenic and preserves the initial diploid genome content in laboratory strains[20]. Furthermore, the LOHs induced by RTG can lead to rapid phenotypic diversification through the unmasking of beneficial alleles[20,21]. Given that RTG induces phenotypic variation without complete sexual reproduction, which often is defective in industrial strains, RTG may represent a powerful approach for improvement of industrial strains. However, it is still unknown whether the RTG paradigm can induce recombination and unlock novel phenotypic variability without triggering systemic genomic instability in more complex genomic scenarios. Furthermore, the selection of RTG recombinant clones either requires genetically engineered selective markers[20] or a lengthy and low throughput microdissection approach[21]. This hinders the scaling-up of the RTG process to generate a large library of recombined industrial yeasts.

In this work, we develop a comprehensive workflow to improve polyploid industrial yeasts through RTG and easily select RTG-recombined clones. We apply this workflow to generate a non-GMO library of evolved industrial polyploid strains harbouring improved biotechnological traits.

## Results

**Genomic and reproductive portraits of industrial polyploid yeasts.** Domesticated strains derived from wild yeast ancestors show hallmarks of genomic complexity such as polyploidy, aneuploidy and horizontal gene transfer (HGT) (Fig. 1a). In order to develop a tailored RTG framework for industrial strains (Fig. 1b), we selected two genetically unrelated industrial *Saccharomyces cerevisiae* strains as test cases, hereafter called OS1364 and OS1431 (Fig. 1a), and characterise their genomes and reproductive capacity. OS1364 was isolated from a cassava factory in Brazil and belongs to the mosaic beer clade, whereas OS1431 was isolated from an ale beer fermentation in England and belongs to the ale beer clade[16] (Supplementary Data 1). We performed both short- and long-reads sequencing and detected multiple hallmarks of domestication. Despite their genetic ancestry, both strains are polyploid, with OS1364 being triploid (3n, with one additional copy of chromosome III) and OS1431 being tetraploid (4n) (Supplementary Figure 1a, b). Furthermore, both genomes harbour a considerable number of heterozygous positions (~40 K) distributed genome-wide consistent with these two strains are intraspecies hybrids generated by admixture of different *S. cerevisiae* lineages (Fig. 1c, Supplementary Figure 1c). A large region of loss-of-heterozygosity (LOH) is present on chromosome XII downstream of the rDNA locus in both strains, although in OS1431 the LOH region is preceded by a heterozygous region. (Supplementary Figure 1c). This is consistent with the rDNA locus being inherently prone to recombination[5,16]. We detected several single-nucleotide polymorphisms within coding regions ($n = 28932$ OS1364, $n = 32399$ OS1431) with OS1431 harbouring more than the expected number (expected ~75%, based on the reference genome, observed 82.2%). Among these polymorphisms we found pervasive heterozygous missense variants ($n = 10747$ OS1364, $n = 13033$ OS1431, Supplementary Figure 1d). We observed that the genetic variation also manifests in the form of copy number variants (CNVs) (Supplementary Figure 1b), resulting in small amplifications or deletions similar to those previously observed in industrial strains[5]. Subtelomeric regions are known to be highly enriched in this type of variation[25]. For instance, we detected a large HGT region close to the subtelomere XIII-R of OS1364 (present in two homologs) harbouring genes derived from *Zygosaccharomyces*

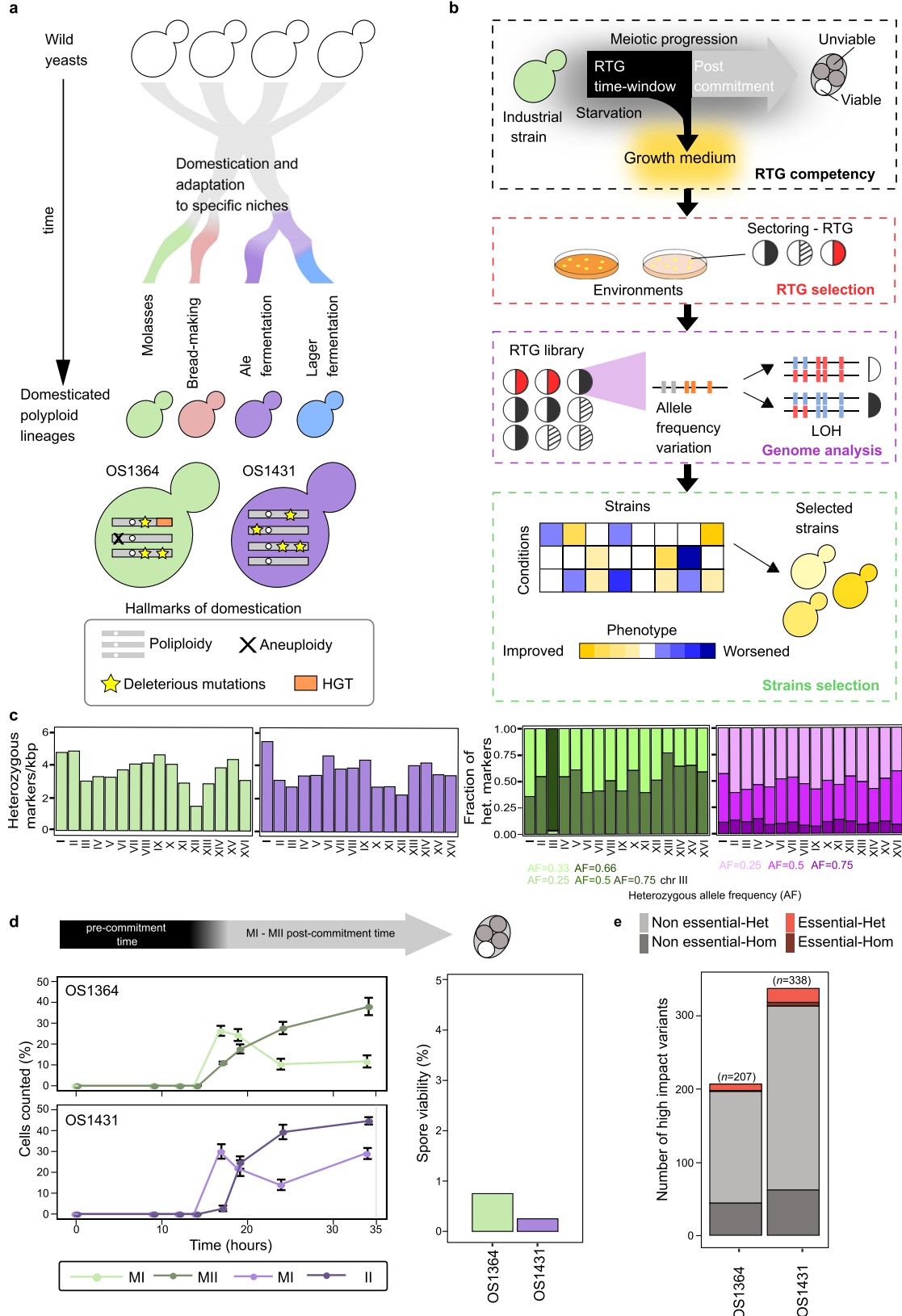

*bailii* (Supplementary Figure 1e). This HGT region has been previously described in wine strains and is known to harbour genes that improve fitness in the fermentation environment[26]. The genetic diversity observed in the two strains reflect the different evolutionary histories of their haplotypes before or after the admixture.

Next, we investigated the reproductive capacity of these polyploid strains and observed a defective sexual reproduction (Fig. 1d). Specifically, the strains showed slow, asynchronous and inefficient meiotic progression and mostly generated unviable gametes (<1%). The high heterozygosity and lack of an effective sexual reproduction suggest that the genetic load

**Fig. 1 Genome complexity and sterility of domesticated *S. cerevisiae*. a** Polyploid strains are characterised by common genomic hallmarks (bottom box), which are detected in both strains used in this study. **b** Schematic depicting the RTG framework: we initially quantify and optimise the meiotic progression, followed by selection of novel colony phenotypes. Finally, RTG samples are sequenced and phenotyped to identify clones with improved industrial traits. **c** Left: level of heterozygosity (number of heterozygous markers/kbp) across chromosomes, total number of heterozygous markers $n = 40431$ OS1364 (green), $n = 39376$ OS1431 (purple). Right: Heterozygous markers partitioned according to the frequency of the non-reference allele. **d** Meiotic progression measured by nuclei DAPI staining, reported as the average ($n = 3$ replicates) of the percentage of cells that have progressed over the first (MI) and second (MII) meiotic division. The error bars represent the standard deviation. On the right, bar plot representing the spore viability (three viable spores/400 OS1364, one viable spore/400 OS1431). The top arrow is a qualitative description of the meiotic progression based on the results of the DAPI staining. **e** The bar plot represents the number of high-impact variants affecting essential (red) or non-essential (grey) genes and further divided into homozygous (Hom) if present in all the haplotypes, and heterozygous (Het) if not present in all the haplotypes. Source data are provided as a Source Data file.

might contribute to gametes' unviability. By using Ensembl variant effect predictor (VEP) (Methods), we detected many highly deleterious variants in both genomes (start-loss, stop-loss, stop-gain, $n = 207$ OS1364, $n = 338$ OS1431), including 10 (OS1364) and 24 (OS1431) in genes that are essential in S288C[27] (Fig. 1e, Supplementary Data 2 and 3, Methods). OS1364 has accumulated a number of deleterious variants not significantly different from the average number of LOFs observed in domesticated strains of the 1011 *S. cerevisiae* collection[13], whereas OS1431 has accumulated significantly more LOFs (Supplementary Figure 1f). Given OS1431 is tetraploid and tetraploid strains correctly segregate their chromosomes in contrast to triploid strains[28], the observed genetic load might contribute to its sterility. Furthermore, we performed a GO-term analysis to test if genes involved in the sporulation process had accumulated LOFs and found no enrichment consistently with both strains entering meiosis when starved. Overall, these two sterile domesticated *S. cerevisiae* strains with complex polyploid genomes represent ideal test cases to probe the RTG framework.

**Sterile polyploids are RTG competent**. Despite the fact these polyploid *S. cerevisiae* hybrids show an inefficient meiotic progression and generate unviable gametes, we conjectured that RTG should not be precluded since it only relies on the meiotic prophase progression, a time window in which cells are not yet committed to complete meiosis. To test if our two polyploid hybrids can perform RTG, we engineered them with a heteroallelic *LYS2/URA3* genetic system to measure LOH rates at the *LYS2* locus on chromosome II, that we have broadly applied in RTG experiments[20,29] (Fig. 2a). We validated the genotype of the engineered *LYS2/URA3* strains by PCR and growth on selective media (Supplementary Figure 2a, Supplementary Data 4). Then, we evolved the engineered OS1364$^{LYS2/URA3}$ and OS1431$^{LYS2/URA3}$ through RTG and collected cell populations throughout the meiotic progression (Fig. 2b). We calculated the RTG-induced recombination by comparing the basal level of cells growing on 5-FOA measured in the unsporulated cultures (T0) to the RTG cells obtained after 6 hours (T6) and 14 hours (T14) of sporulation induction (Methods). We detected a 10-fold and threefold increase of cells growing on 5-FOA, indicating an increased LOH rate at T14 in OS1364$^{LYS2/URA3}$ and in OS1431$^{LYS2/URA3}$, respectively, whereas the increase was not significant at T6, consistent with the fact that cells did not progress sufficiently through meiosis at this timepoint (Fig. 2b, Supplementary Data 5). The absolute percentage of cells grown at T14, calculated subtracting the T0 background value to the T14 (Methods), was almost twofold higher in OS1431$^{LYS2/URA3}$ (0.10% in OS1364$^{LYS2/URA3}$, 0.19% OS1431$^{LYS2/URA3}$), in line with its lower heterozygosity on chromosome II (Fig. 1c) where pre-existing small LOH regions might represent preferential sites of recombination[29]. To further prove that RTG induced the increase in recombination observed at the *LYS2/URA3* locus we deleted all three copies of *SPO11*, a gene essential for inducing DSBs in meiosis, in OS1364$^{LYS2/URA3}$ (Supplementary Figure 2b) and measured

recombination. We did not detect any significant increase between the T0 and the respective T14 in the OS1364$^{LYS2/URA3}$ *spo11Δ* strain (Supplementary Figure 2c, Supplementary Data 5), consistent with RTG caused the increased recombination as it relies on the Spo11p induced DSBs in early meiosis.

We next performed whole-genome sequencing of the *LYS2/URA3* parental strains, controls (T0) ($n = 2$ OS1364, $n = 2$ OS1431, Supplementary Data 1) and RTG-evolved clones (T14) ($n = 11$ OS1364, $n = 11$ OS1431, Supplementary Data 1) isolated on 5-FOA plates to evaluate the genome-wide impact of RTG. Our analyses revealed varying levels of recombination in the RTG clones, consistent with previous reports in RTG diploid hybrids[20,21] (Fig. 2c, d, Supplementary Figure 2d). The RTG clones derived from OS1364$^{LYS2/URA3}$ had an average of 10% (maximal value 26%) of the genome affected by LOH, whereas OS1431$^{LYS2/URA3}$ had an average of 6.7% (maximal value 13.5%). Moreover, we did not detect any chromosome loss potentially accounting for 5-FOA resistance in the RTG sequenced clones, showing that despite the strain polyploidy, LOH at the *LYS2/URA3* locus arose more frequently than aneuploidy during RTG. We observed overall genome-wide stability and detected only one aneuploidy and two large CNVs across the 11 RTGs derived from OS1431$^{LYS2/URA3}$, where the two CNVs likely resulted from ectopic recombination initiated by a small region of homology between chromosome IX and chromosome XIV (Supplementary Figure 3a–c). Similarly, we detected only two aneuploidies in one RTG sample across the 11 samples derived from OS1364$^{LYS2/URA3}$. Altogether, these results demonstrate that these polyploid hybrids are RTG-competent despite their meiotic sterility, supporting that the RTG workflow can generate genetic diversity in sterile industrial strains.

**Selection of recombined RTG clones by natural colony phenotypes**. The RTG selection based on *URA3*-loss requires genome editing and produces GMOs that have marketing constraints in the food industry. To overcome this limitation, we devised an alternative selection strategy based on unmasking natural variability in colony phenotypes such as morphology and colour. Colony phenotypes are complex traits, influenced by both genetic and environmental factors (Fig. 3a). We used a YPD-based medium with varying concentrations of dextrose (0.5% and 1%), a major environmental regulator of colony morphology[30,31], to unveil phenotype of colonies derived from RTGs (Fig. 3a, Methods). Indeed, we found phenotypic variability in the RTG colonies but the divergent phenotype was often present in only a large sector (approximately half) of the colony. Therefore, we hypothesized that the sectored colonies were mother-daughter (M-D) RTG pairs that completed bud separation after plating and then growing to form a bi-sector colony. Consistently, we found sectored and non-sectored colonies according to when cells were plated after the RTG induction (Supplementary Figure 4a). We observed three types of sectored colonies in the RTG plates of OS1364 (wrinkled-smooth, red-white and dark-white) (Fig. 3b, Supplementary Figure 4, Supplementary Data 5) with a summed

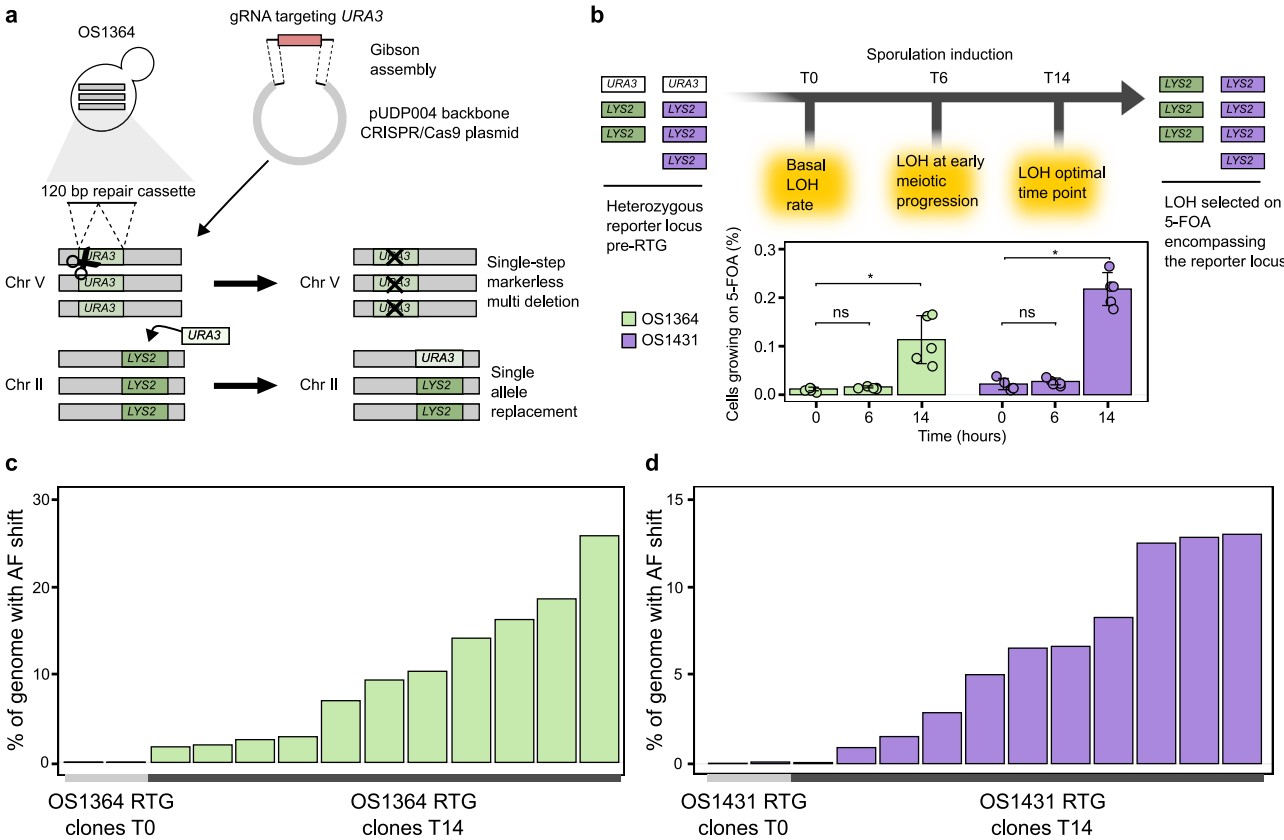

**Fig. 2 RTG recombination in sterile polyploids. a** A *URA3*-loss genetic system in the polyploid strains was engineered with a CRISPR-based approach. The strains generated in such way are hereafter referred to as *LYS2/URA3* regardless of the number of loci. **b** The *URA3*-loss genetic system was used to measure recombination rate at the heterozygotic locus *LYS2/URA3* (top panel). Bar plots representing the average (*n* = 5 replicates) percentage of cells growing on 5-FOA at different time points: no sporulation induction (T0), 6 hours (T6) and 14 hours (T14) of sporulation induction (bottom panel). The increase of cells growing on 5-FOA at T14 compared to T0 was significant for both samples (*p* value = 0.006., one-sided Wilcoxon ranked-sum test) whereas was not significant at T6 (*p*-value = 0.06, one-sided Wilcoxon ranked-sum test). The error bars represent the standard deviation. **c, d** Bar plot representing the percentage of genome in which we detected an allele frequency (AF) shift for the RTG samples selected for *URA3*-loss derived from OS1364^LYS2/URA3 (green, **c**) and OS1431^LYS2/URA3 (purple, **d**). Recombination and LOH induced upon genome engineering with CRISPR/Cas9 at the native *URA3* locus was excluded. Source data are provided as a Source Data file.

total frequency of ~0.75% and one type of sectored colony in the OS1431 RTGs (wrinkled-smooth) with a frequency of ~0.5% across all environments tested (Fig. 3b, Supplementary Data 5). In contrast, we did not observe any of these phenotypes in control plates where cells were plated without inducing sporulation. Thus, this result was in line with our initial hypothesis that the sectored colonies were unveiled in the specific media upon budding of recombined mother-daughter (M-D) RTGs. In contrast, the non-sectored colonies displaying phenotype variation represent cells in which the RTG mother and daughter were separated before plating. An alternative explanation is that the sectored RTG colonies derived from residual sporulation and spore germination, although this is unlikely given the near to zero spore viability observed. To undoubtedly exclude this scenario, we deleted the *NDT80* gene by CRISPR/Cas9 multi-deletion in both hybrids, to generate mutants that are RTG competent[20,24] but arrest before the first meiotic division (MI) and therefore unable to complete sporulation (Supplementary Figure 5a, b). Then, we evolved the OS1364^ndt80Δ and OS1431^ndt80Δ mutants through the RTG protocol (Supplementary Figure 5b) and detected sectored phenotypes similar to those seen in the WT hybrids, thus excluding the possibility that residual sporulation contributes to the formation of sectored colonies.

We sequenced multiple paired samples derived from sectored RTG colonies and also single non-sectored RTG colonies derived

from RTG in the OS1364 and OS1341 wild-type and *ndt80Δ* strains and samples from control plates (T0) (Supplementary Data 1). Whole-genome sequencing revealed recombination detected as allele frequency (AF) shifts across the unphased heterozygous markers in all the putative M-D RTG colonies with a sectored phenotype (Fig. 3c, d, Supplementary Figure 5c, d). The fraction of the genome in which we detected RTG-induced recombination was highly variable in both strains (9.7 ± 8.2% in OS1364, 4.5 ± 2.7% in OS1431 WT-RTG, average ± SD), mirroring the results obtained from the *URA3*-loss assay. Moreover, we provided two additional cues that the sectored RTG colonies were genuine RTG M-D pairs. First, the AF shift for heterozygous markers was largely reciprocal in the wild-type and *ndt80Δ* sectored RTG pairs with the rest of the non-reciprocal events representing gene-conversion (Fig. 3e, Supplementary Figure 6a–c, Supplementary Data 12) as found in M-D RTG pairs of diploid lab strains. In addition, by comparing each pair-wise combination of RTG M-D pairs we also found that each pair is clearly differentiated from the others in terms of recombination landscape (Supplementary Figure 6d).

Some of these regions of recombination were shared across RTG M-D pairs of OS1364 with the same sectored phenotype (white/darker) (Supplementary Figure 7a) supporting a localised LOH underlying the colony phenotype. However, owing to the complex nature of the wrinkled-smooth phenotype we could not

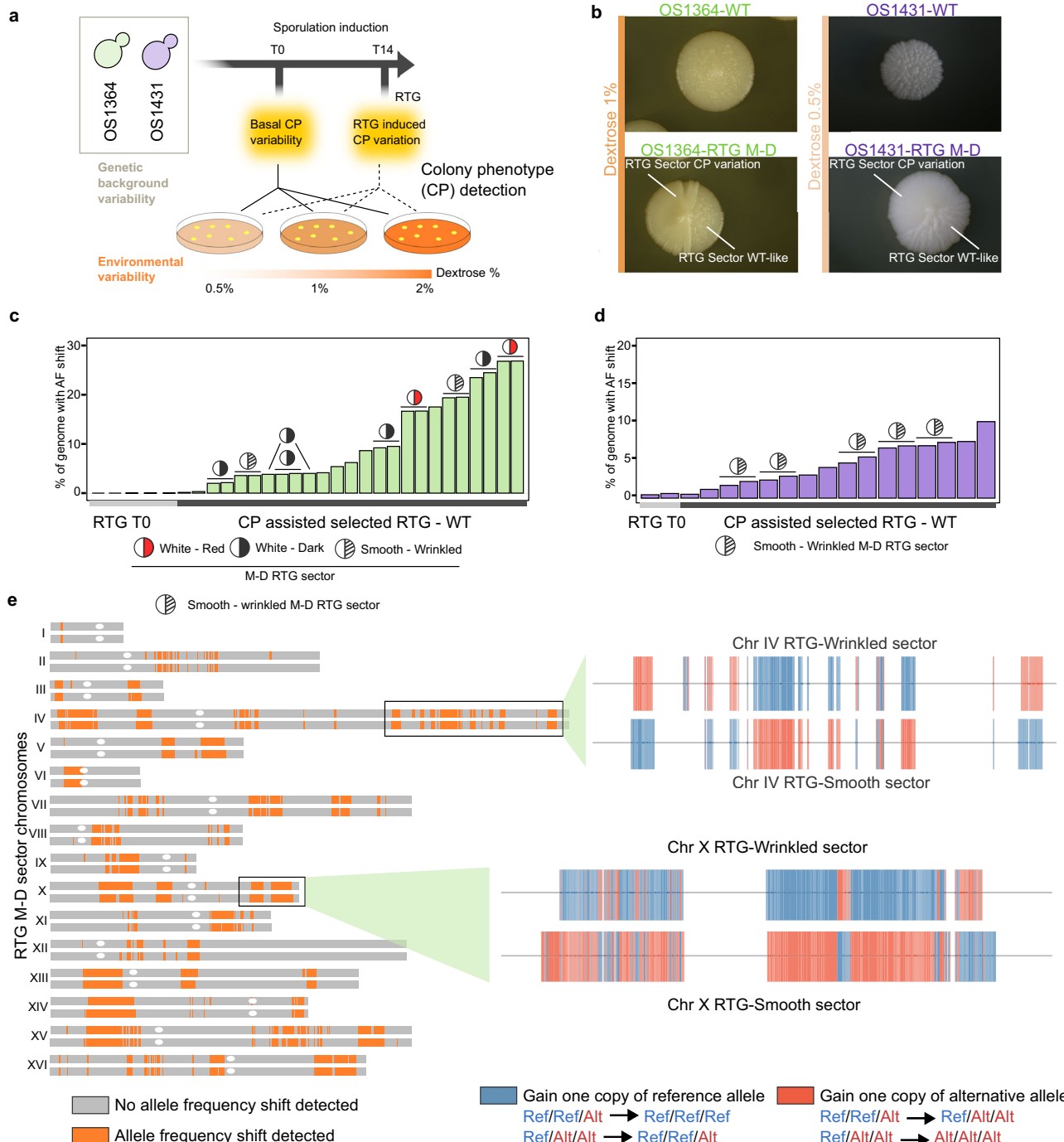

**Fig. 3 Colony phenotyps revealed mother-daughter RTG pairs.** **a** Varying dextrose concentrations unveiled hidden colony phenotypic variation upon RTG. **b** Wild-type (WT) colony phenotypes (top) and sectored phenotypes emerging on RTG plates (bottom). The concentration of dextrose in the media is reported on the left. **c** The percentage of markers with allele frequency (AF) shift in the OS1364 derived RTG samples. The mother-daughter pairs for which both sectors were sequenced are indicated (circles). Single samples represent non-sectored RTG colonies (n = 4) or only one-side of a RTG sectored colony (n = 3). **d** As in **c** for the OS1431 background. **e** Recombination map of a RTG M-D pair (top chromosome: sm244/ bottom chromosome: sm245, derived from OS1364). Grey regions indicate the heterozygous markers without AF shifts, whereas orange regions contain heterozygous markers with AF shifts underlying recombination events. A zoom-in of two recombined regions is illustrated on the right. Colours represent the changes in genotype ratio of heterozygous markers. The lack of completely phased haplotype generates a fragmented AF shift pattern although this might derive from a longer continuous recombination event (Methods). Source data are provided as a Source Data file.

find any shared AF shift region among the respective RTGs derived from OS1364 or OS1431. Second, we observed a complementary gain and loss of one chromosome in one M-D pair derived from OS1364 (Supplementary Figure 7b), which is compatible with chromosome missegregation between M-D

during RTG. Thus, we concluded that the sectored RTG colonies were indeed M-D pairs and we exploited the recombination generated by RTG in OS1364 M-D RTGs to produce phased local haplotypes showing a potential application of RTG in polyploid yeasts (Supplementary Figure 7c).

Overall, we showed that highly recombined RTG M-D pairs can be selected by exploiting natural colony phenotypic variation and proved that these phenotypes arise as a result of RTG-induced recombination.

**RTG recombination shapes industrial fitness.** Next, we probed the potential of RTG to improve industrial phenotypes. First, we compared the fermentation performances of OS1364 and OS1431 to the commercial diploid strain WLP001 to evaluate the fermentation performances of our parental strains (Supplementary Data 1). We confirmed that our strains are competitive for industrial fermentation, with OS1364 outperforming both WLP001 and OS1431 for fast fermentation (Supplementary Data 10). However, both OS1364 and OS1431 had a substantial decline in cell viability (of ~25% and ~50%, respectively) often observed in chronologically aged polyploid *S. cerevisiae*[13], which is more modest in WLP001 (8%, Supplementary Data 11).

We screened selected non-GMO RTGs ($n = 25$ OS1364, $n = 16$ OS1431, Supplementary Data 6) and compared them to their respective parental strain and T0 controls ($n = 2$ OS1364, $n = 2$ OS1431) across several conditions mirroring industrial fermentations (Fig. 4a). We screened osmotic and alcoholic stress conditions and found that RTG samples had broad phenotypic variability with either a worsened, unchanged or improved phenotype compared to parental strains and T0 controls (Fig. 4b, Supplementary Data 6). This scenario is consistent with the LOHs induced by RTG arising randomly in the genome and not as a by-product of a specific selective pressure. Moreover, some M-D RTG pairs showed complementary growth-rate variation (Fig. 4c) that can be explained by complementary LOHs segregating weaker and stronger alleles in the RTG pair. We binned the M-D RTG pairs according to the two colony morphology phenotypes for which we had at least 5 RTG pairs (OS1364: dark/white, OS1431: smooth/wrinkled) to test co-segregation with industrial traits. We observed a significant difference in growth rate in maltose and sorbitol of RTG M-D pairs derived respectively from OS1364 and OS1431 (Supplementary Figure 8a) suggesting that specific traits can partially co-segregate with the colony phenotype selected. Finally, we found that the growth-rate variation in ethanol and maltose appeared to be moderately correlated in the OS1364 RTG library, underscoring that recombination might have affected pleiotropic genes regulating both traits (Supplementary Figures 8b, c).

Next, we selected RTG samples derived from OS1364 based on their phenotypic performance ($n = 12$) and their LOH landscape across a core set of fermentation genes ($n$ genes = 74), and all the M-D RTG pairs from OS1431 ($n = 10$) (Supplementary Data 7 and 8). We inoculated the selected RTGs in a high-gravity wort (32° Plato), representing extreme fermentative conditions, to further investigate differences in stress resistance, carrying out a flask-scale fermentation for 13 days evaluating mass loss and ethanol produced at regular intervals (Supplementary Data 9). Some RTG samples showed increased fermentation kinetics (Fig. 4d) as well as a superior alcohol production (Supplementary Figure 8d, e) compared to the parental strains. We also noticed that in some M-D RTGs derived from OS1431 the fermentation performances were complementary (Supplementary Figure 8e, Supplementary Table 9). We hypothesized that RTG samples showing improved fermentation performances should also have improved resistance to fermentation stress and, potentially, higher post-fermentation cell viability. Therefore, we selected two RTG variants for each background among the best performers in the previous flask-scale fermentation (Supplementary Figure 8c, d, Supplementary Data 9), to carry on a two-liter scale fermentation in high-gravity wort, to better mimic modern

industrial brewery practice (20° Plato). We found that the OS1364 RTGs performed at least equally well as their respective parental strain reaching a similar level of alcohol by volume at the end of the fermentation although one RTG showed a slower fermentation kinetic between 3 and 6 days (Supplementary Figure 9a, Supplementary Data 10). Remarkably, one OS1364 RTG had a large increase in post-fermentation cell viability (Fig. 4e, Supplementary Data 11), reaching a level similar to WLP001, showing that RTG can purge detrimental traits. In contrast, OS1431 RTGs were slightly better compared to their parent in fermentation performances (Supplementary Figure 9a, Supplementary Data 10) but without increased post-fermentation viability (Fig. 4e).

Furthermore, we evaluated the aroma profile and the post-fermentation residual sugars, two parameters that are highly relevant in the beer industry. The two parental strains produced distinct aroma profiles from that of WLP001 and consumed almost all the sugars present in the wort (Supplementary Data 11). The four selected RTG samples did not show any deleterious trade-off in either phenotypes with the exception of a slight increase of acetaldehyde, in sm408, which is not desirable but its increase was negligible as those sensory thresholds can be individual dependent and affected by the beer style (Sensory threshold = 10 mg/mL, sm408 = 10.5 mg/mL, WLP001 = 21.47 mg/mL, Fig. 4f, Supplementary Data 11). One RTG derived from OS1364 had lower production of ethyl acetate (Fig. 4f, Supplementary Figure 9c, Supplementary Data 11). This is consistent with recombination encompassing the gene *ATF2*, which has a role in shaping this trait (Supplementary Data 8). Moreover, the RTG derived from OS1364 showed diversified production of esters compared to the respective parental strain, whereas this was limited to only 2-Phenylethylacetate for the RTG from OS1431 (Fig. 4f). This variability may lead to further differentiation of the sensory profile of the beer produced by the RTG variants.

Overall, our data demonstrate that RTG recombination in sterile polyploid strains can unlock phenotypic variability in traits of industrial relevance[19], contributing toward microbial stability or shaping the sensory quality of the product.

## Discussion
In this work, we showed that RTG is an efficient approach to generate genetic and phenotypic diversity in two fundamentally different industrial sterile yeast strains. Current approaches to improve industrial yeasts largely depend on designed targeted genetic modifications and must contend with market restrictions and societal mistrust[19]. As an alternative, we now show that RTG-assisted homologous recombination is an efficient approach to generate genetic and phenotypic diversity in polyploid sterile industrial yeasts, which extends its developments in diploid artificial hybrids that generates phenotypic variability across a wide range of conditions[20,21]. The RTG, coupled with selection based on natural phenotypes generates non-GMO yeast strains that can be unrestrictedly introduced into the market. Compared to other non-GMO approaches such as serial transfer[32,33], RTG induces genome shuffling of multiple loci and it does not select for a specific trait except for the genetic loci regulating the selected colony phenotype. The genome shuffling occurs in association with meiotic hotspots and does not select specific parental alleles[20]. Therefore, any region of the genome can virtually experience recombination mediated by RTG with a reduced frequency in cold regions such as centromeres. However, possible constraints due to the unmasking of recessive deleterious alleles may contribute to further shaping the LOH recombination landscape. Nevertheless, RTG has several benefits over the previous improvement strategies. We showed that RTG does not

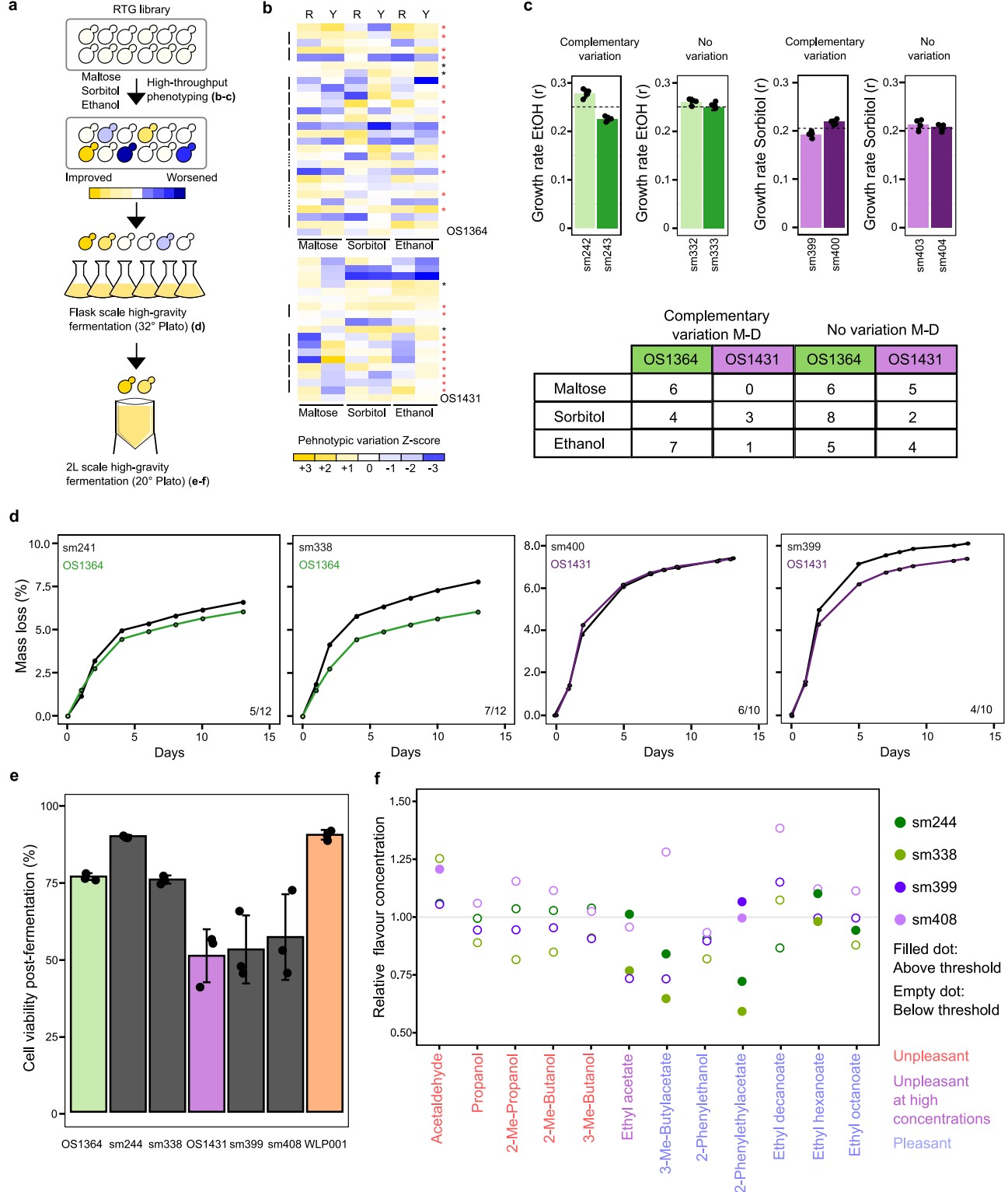

trigger genome instability in polyploid strains, similarly to diploids[20,21]. This is in contrast with direct selection where the abundant number of generations performed often results in ploidy and chromosome copy number variations[34]. Furthermore, trade-offs in unselected traits have been described as undesirable outcomes in adaptive evolution experiments[35–38], although some solutions have been proposed to fix, for instance, the impaired sugar consumption emerging after adaptive evolution[38]. We also observed RTG variants with impaired performances. In fact, we

found that several RTG M-D pairs had complementary worsened and improved fitness in different environments, in line with what was observed in M-D RTG pairs derived from a diploid hybrid[21]. However, it is possible that undesired trade-offs may be selected with the colony morphology phenotype selected. For instance, the wrinkled RTG sectors derived from OS1431 exhibited a lower growth rate in sorbitol compared to that of smooth RTG sectors. The selection of M-D pairs may enable to mitigate this scenario by picking-up only sectors of the M-D pairs in which the colony

**Fig. 4 Phenotypic variability in industrial traits. a** Schematic of the phenotypic screening performed. Letters in parentheses indicate the respective figure panels where data are reported. **b** Heatmaps representing growth rate (1st column, R) and yield (2nd column, Y) of the RTG samples selected for the next screening (red star), T0 samples (black star) and the M-D sectors (connected with lateral black line) and RTG-sectors for which only one sector was sequenced (dotted-black line). Each measure is the average of 4 technical replicates. **c** Phenotypic diversification in sectored M-D pairs of OS1364 (green) and OS1431 (purple). Bar plots represent examples of RTG M-D pair with either divergent phenotypic variation or absence of variation. The dotted line represents the average phenotype in the RTG M-D library. Each bar plot represents the mean of $n = 4$ independent replicates and the error bar represents the standard deviation. Bottom table reports the number of M-D pairs showing a complementary phenotypic variation. **d** Mass loss curves during the flask-scale fermentation experiment of an RTG without variation (left) and an improved RTG (right) for OS1364 (green) and OS1431 (purple). Each point represent the average of $n = 2$ independent fermentations. The numbers on the bottom right represent the fraction of RTG samples with similar trends. **e** Post-fermentation viability measured after a 2L-fermentation in high-gravity wort for the two parents, the respective RTG samples and a commercial strain in triplicate. One RTG (sm244) derived from OS1364 showed a huge increase in viability. Each bar plot represent the average of $n = 3$ independent fermentations and the error bars represent the standard deviation. **f** Variability in the aroma profile across the four RTGs from the 2 L scale fermentation, the values are expressed as relative change compared to the concentration produced by the parental strain. Empty and closed dots respectively represent compounds with concentrations below or above the sensory threshold. Source data are provided as a Source Data file.

phenotype is not associated with a worsened fitness. Multiple deleterious trade-offs can accumulate in RTG samples when the fraction of the genome that recombines is large. However, the sequenced RTG samples have variable levels of genome-wide recombination, and samples with a low level of recombination might be less likely to experience undesired trade-offs. Nevertheless, one of the most recombined RTG did not show phenotypic decay in the screenings performed; on the contrary, it was better able to tolerate the harsh fermentation conditions and had increased post-fermentative viability.

With the present approach, the RTG libraries are produced without introducing any selective pressure. Therefore, they are phenotypically agnostic, except for the trait used for the colony phenotype selection, and can harbor RTG variants with improved phenotypes that are useful for multiple applications. The variability of colony-associated phenotypes is a key parameter of the approach we devised for selecting M-D RTG recombinants, and it might be limited by the variability of natural phenotypes. Screening of several environments could provide an effective solution to unveiling variability in colony phenotypes. For instance, there are media that can trigger a colour variation linked to relevant industrial traits, such as the production of hydrogen sulfide detected in BiGGY agar plates[39]. Moreover, our approach successfully captured mother and daughter RTG pairs that have only been isolated so far from laboratory strains with a micro-manipulation approach[21] that cannot be scaled up. We envision that our selection method can be automated to quickly screen hundreds of colonies using, for instance, high-throughput approaches[40]. In addition, our selection protocol can be easily performed and seamlessly integrated with other RTG methods that do not feature mother-daughter RTG pairs[41] or other yeast improvement strategies. For instance, strains derived from an adaptive evolution experiment could be further improved through RTG to erase trade-offs, or, vice-versa, RTG samples could be submitted to adaptive evolution to be further optimized. The RTG framework can be generalized to reshuffle designed hybrid genomes[20,21] as the only requirement is that the hybrid can enter meiosis and progress until the first meiotic prophase, even if with low efficiency. One limitation is that RTG does not allow bringing to the designed hybrid further genetic variability by breeding and thus is especially suited for sterile strains. However, we showed that diploid hybrids that experience recombination at the *MAT* locus can become maters and generate polyploid hybrids[17]. This mechanism could be exploited to breed sterile hybrids with gametes derived from a diverse genetic background and could be used together with iHyPr[9] and multi-generational breeding[8] in yeast improvement programs. The RTG approach has other practical applications as RTG libraries can be used in large-scale linkage analyses to unravel the genetic

architecture of complex industrial traits and the RTG induced recombination could aid existing genome phasing methods[42] in producing polyploid *de novo* genome assemblies. In conclusion, we propose the RTG framework as an avenue to induce genetic and phenotypic variability in yeast strains, including industrial sterile yeasts, and can also provide insights into complex genetic traits in polyploid strains.

## Methods

**Detection of deleterious, missense mutations, and pre-existing LOHs**. Short reads of the parental strains were obtained from the 1011 *S. cerevisiae* yeast project[16] and mapped against the SGD reference R64.2.1 using bwa-mem algorithm. Single-nucleotide variants were called by using Freebayes (v1.3.1–19) with the argument "-p" to set an appropriate ploidy and quality >20. We annotated impactful mutations by using the VEP suite[43]. Mutations were annotated as impactful if they caused a stop-loss, stop-gain, or start-loss. Subtelomeric variants or frameshift ones were included in the list to make the data comparable to the table of LOFs generated for the 1011 *S. cerevisiae* yeast project[16]. The list of essential genes was obtained from Liu et al[23]. It is possible that some genes listed as essential are such because their deletion or alteration affects the gene on the opposite strand. We found three such cases in the essential genes detected as homozygous, which do not generate LOF on the gene on the opposite strand, so their essentiality may not be as predicted in Liu et al.[27]. The GO-term analysis was performed using the SGD GO-term analysis suite at https://www.yeastgenome.org/ with a *p* value threshold of 0.01 and selecting GO-term associated with sporulation. We identified pre-existing LOHs in the genome of the parental strains by searching for non-overlapping regions of 50 kbp where we identified 10 or less heterozygous markers that were not shared by all the unphased haplotypes. The plots were generated using ggplot2 and in-house R scripts.

**Genome content analysis**. The genome content of OS1364 and OS1431 was measured using propidium iodide staining. Each strain was patched from a −80 °C glycerol stock onto a YPD plate (1% yeast extract, 2% peptone, 2% dextrose, 2% agar) and incubated overnight at 30 °C. The following days the strains were incubated in 1 mL liquid YPD medium (1% yeast extract, 2% peptone, 2% dextrose) and grown overnight at 30 °C without shaking. The next day 200 μL of the overnight culture was resuspended in 1 mL of fresh YPD and growth until the exponential phase. Then cells were centrifugated, washed with 1 mL water, and fixed overnight in 1 mL of EtOH 70%. The following day each sample was washed with PBS 1×, resuspended in the PI staining solution (15 μM PI, 100 μg/mL RNase A, 0.1% v/v Triton-X, in PBS), and incubated for 3 hours at 37 °C in the dark. Ten thousand cells for each sample were analysed on a FACS-Calibur flow cytometer. Cells were excited at 488 nm and fluorescence was collected with an FL2-A filter. An increase of 50% of the fluorescence value (a.u.) compared to the fluorescence signal of the diploid cells in G1 was used to assign a ploidy of 3n. An increase of 100% of the fluorescence value (a.u.) compared to the fluorescence signal of the diploid cells was used to assign a ploidy of 4n.

**Long-read sequencing and HGT identification**. Yeast cells were grown overnight in a liquid YPD medium. Genomic DNA was extracted using Qiagen Genomic-Tips 100/G according to the manufacturer's instructions. The MINION sequencing library was prepared using the SQK-LSK108 sequencing kit according to the manufacturer's protocol. The library was loaded onto an FLO-MIN106 flow cell and sequencing was run for 72 hours. We performed long-read basecalling and scaffolding using the pipeline LRSDAY[44]. The Canu assembler mostly merged the different haplotypes and thus prevented to production of long-read phased genomes. The dotplots were generated by using nucmer and mummerplot[45].

The annotated non-reference regions on chromosome XIII of OS1364 were extracted from the fasta file of the collapsed assembly genomes (the haplotypes were not phased) and blasted by using the application "blastn" at https://www.ncbi.nlm.nih.gov/ against the database of ascomycetes.

**Sporulation monitoring and spore viability**. Yeast strains were patched on YPD plates and incubated at 30 °C for 24 hours. From the patch a streak was done and incubated at 30 °C for 48 hours, then single colonies were isolated and grown overnight at 30 °C in 10 mL of liquid YPD in a shaking incubator at 220 rpm. The following day, single colonies were inoculated in different tubes containing 10 mL of the pre-sporulation medium SPS (1% peptone, 1% potassium acetate, 0.5% yeast extract, 0.17% yeast nitrogen base, 0.5% ammonium sulfate, 1.02% potassium biphthalate) and kept at 30 °C in a shaking incubator for 24 hours. Tubes were then centrifuged and washed three times with sterile water and cells were resuspended in erlenmeyer flasks containing 25 mL of KAc 2% to reach a final OD of 1, and incubated at 23 °C in a shaking incubator. Sporulation was monitored by DAPI staining of the sporulation cultures to detect cells that have passed the first meiotic division, and contain two nuclei, or the second meiotic division, and contain four nuclei. The cells taken from the meiotic culture were first washed with 1 mL water and then fixed with 1 mL EtOH 70% overnight. Then, 200 μL of culture were stained with 2 μL of DAPI (4′,6-diamidino-2-phenylindole) 0.5 μg/mL, and 10 μL of stained cells were spread on a microscope slide and incubated in the dark for 40 minutes. To infer the efficiency of meiotic progression the cells were examined by fluorescence microscopy (Imager Z1 Zeiss) using a DAPI filter and scored as having one, two, or four nuclei, which indicates if they have not progressed after MI (one nucleus) or have progressed after MI (two nuclei) or MII (four nuclei). For assessing the spore viability, the spores were collected from the sporulated cultures and incubated between 30–60 minutes in 100 μL of zymolase solution in order to perform spore dissection. At least 100 tetrads per sample were dissected on YPD plates using a SporePlay(+) (Singer). Spore viability was assessed as the number of spores forming visible colonies after 4 days of incubation at 30 °C.

**Plasmid engineering and genome editing with CRISPR/Cas9**. The multi-deletions of *URA3*, *NDT80*, and *SPO11* were engineered by using CRISPR/Cas9 genome editing. The plasmid harboring Cas9 was obtained from Addgene pUDP004[46] and linearized with BsaI. The resistance to acetamide was replaced with the resistance cassette to Kanamycin which was amplified from a plasmid harboring the KanMX resistance cassette. The gRNA with the necessary nucleotide for self-cleavage was designed on UGENE[47] and ordered as a synthetic oligo from Eurofins Genomics (™). The synthetic oligo was cloned into the plasmid backbone by using the Gibson assembly kit (NEB, Gibson Assembly®) and the ligation reaction was carried out for 1 hour at 50 °C. The assembled plasmid was transformed into DH5-alpha competent bacteria by heat shock and the bacteria were incubated in 3 mL of LB broth for 1 hour to induce the synthesis of the antibiotic resistance molecules and then plated on LB plates containing 100 μg/μL of ampicillin. The following day, cells were screened by polymerase chain reaction (PCR) using primers to validate the correct golden gate assembly of the construct. Successfully transformed bacterial colonies were inoculated in LB broth containing 100 μg/μL of ampicillin and incubated overnight at 37 °C. Cells were harvested from the overnight incubation and the plasmid was extracted using the QIAprep Spin Miniprep Kit following the manufacturer's instructions.

The 120 bp cassettes used for the deletion of *URA3*, *NDT80*, and *SPO11* were designed to be flanking 60 bp upstream and downstream of the candidate gene, which harbours the cutting site for Cas9 and ordered as a unique synthetic oligo at Eurofins Genomics. The forward and reverse cassettes were mixed at equimolar ratio, heated at 95 °C for 15 mins, and then cooled down at room temperature to be ready for the transformation. Yeast samples were transformed using between 1 and 15 μg of cassette, at least 200 ng of CRISPR/Cas9 plasmid, and following the protocol from[42]. Cells were then plated on selective media containing kanamycin (400 μg/mL) and incubated at 30 °C for 3–7 days. Candidate transformed clones were validated by PCR using primers designed on the outside regions of the deleted genes. Positive clones were streaked on YPD and grown for 2 days at 30 °C to allow plasmid loss. Plasmid loss was confirmed by plating again the colonies in the selective medium and positive ones were patched on YPD and stored at −80 °C in 25% glycerol tubes.

**RTG selection by *URA3*-loss assay**. A *URA3* cassette was introduced in one *LYS2* locus on chromosome II in *URA3* knock-out OS1364 and OS1431 strains by using the classical lithium acetate protocol. The correct insertion was validated by checking for restored prototrophy on synthetic media lacking uracil and by PCR. Positive clones were stored at −80 °C in 25% glycerol tubes. For performing the *URA3*-loss assay, cells from the frozen stocks were patched on YPD plates and incubated at 30 °C for 2 days. Cells were then streaked on plates lacking uracil and incubated at 30 °C for 2 days. Single colonies were picked and sporulation was induced following the protocol described in the paragraph "Sporulation monitoring and spore viability". Cells from the sporulation cultures were taken at 0, 6, and 14 hours after sporulation induction, washed three times with YPD, and incubated in YPD for 12 hours at 30 °C without shaking. Dilutions of the YPD liquid culture were spotted onto YPD plates and an appropriate dilution was chosen for each

strain and plated on 5-FOA (2% dextrose, 0.675% yeast nitrogen base, 0.088% uracil drop-out, 0.005% uracil, 2% agar, 0.1% 5-FOA) plates. The plates were incubated at 30 °C for 2 days and colonies growing on 5-FOA plates were counted at all the time points. We calculated the LOH rate at the three-time points (T0, T6, and T14) as the ratio between the % of cells growing on 5-FOA and the respective percentage of cells growing on YPD, according to the following equations:

$$\text{LOH rate}_{T0} = 100 \times (\text{CFU}_{5-\text{FOA},T0}/\text{CFU}_{YPD,T0}) \quad (1)$$

$$\text{LOH rate}_{T6} = 100 \times (\text{CFU}_{5-\text{FOA},T6}/\text{CFU}_{YPD,T6}) \quad (2)$$

$$\text{LOH rate}_{T14} = 100 \times (\text{CFU}_{5-\text{FOA},T14}/\text{CFU}_{YPD,T14}) \quad (3)$$

where $\text{CFU}_{5-\text{FOA},T0}$ is the number of colony-forming units on 5-FOA at T0 and $\text{CFU}_{YPD,T0}$ is the number of colony-forming units on YPD at T0. These LOH rates were used to calculate: (1) the fold-increase of cells experiencing LOH by dividing the LOH rate at T6 or T14 by the LOH rate at T0 (LOH ratio); (2) the absolute difference of LOH by subtracting the LOH rate at T0 to the respective LOH rates at T6 and T14.

**RTG selection by natural phenotypes**. Sporulation of wild-type hybrids was induced as described in the section "Sporulation monitoring and spore viability" for a time window compatible with RTG (14 hours). Cells were withdrawn from the sporulation medium before the appearance of MI cells (two nuclei) to avoid plating cells committed to complete sporulation. At the appropriate timepoint, cells were shifted from sporulation medium to liquid YPD. Part of the cells was incubated between 2 and 3 hours to induce RTG but not budding and separation of the mother-daughter RTG pairs, while others were incubated longer to allow complete budding and separation of candidates M-D RTG pairs. Then, the samples were plated on modified YPD media (YPD 0.5: 1% yeast extract, 2% peptone, 0.5% dextrose. YPD 1: 1% yeast extract, 2% peptone, 1% dextrose), and plates were incubated at 30 °C and monitored daily for colony formation and morphology variation. M-D RTG pairs were selected as colonies displaying two sectors with different morphology, of which one resembled the wild-type phenotype whereas the other was divergent. Cells from each side of the sectors were taken with a wooden stick and streaked on their respective modified YPD to limit contamination from cells of the other sector, and incubated at 30 °C for 2 days. Then, a single colony was taken, patched on YPD and incubated at 30 °C for 2 days, and finally stored in 25% glycerol tubes at −80 °C. Pictures of the sectoring colonies on the plates were taken with a stereomicroscope Discovery v.8 Zeiss.

**CNVs detection in parental strains and RTG samples**. Short reads Illumina sequencings were performed at the genomic platform of the Institute Curie. Reads were mapped to the S288C reference genome with the bwa-mem algorithm. Optical duplicates of the sequencing were removed using "samtools rmdup". Processed BAM files were then indexed using "samtools index" and coverage was extracted using "samtools depth". Coverage along the chromosomes was plotted using in-house R scripts in which sliding windows of non-overlapping 10 kbp were used to calculate a local average coverage, which was then normalized with the median coverage along the chromosome. Genome-wide coverage profiles were manually inspected to detect artifacts due to smiley pattern[5] or lower coverage affecting only small chromosomes. Aneuploidies and large CNVs were detected by manually inspecting the $\log_2$ coverage profiles, while shorter CNVs (<1 Kbp) were detected by computing the median coverage across sliding windows of non-overlapping 1 Kbp. Regions showing a coverage variation with respect to the median of their respective chromosomes were then matched with the previously detected CNVs.

**Detection of regions with AF shift in RTG samples**. To identify LOHs in the evolved RTGs, we first generated a list of reliable markers represented by all the heterozygous positions between the reference S288C strain and the two polyploid parental genomes (OS1364 and OS1431). Reads mapping and post-processing of sequenced parental strains, T0 and RTG samples were performed as described in the previous paragraph "CNVs detection in parental strains and RTG samples". Variant calling in the parental strains was performed by using Freebayes (v1.3.1–19) with the options "-p" to set the appropriate ploidy. The parental vcf files were then filtered to include only SNP markers with quality >20 and depth >10 using bcftools with the options "TYPE = snp, QUAL > 20, DP > 10". Variant calling on the evolved RTG and control (T0) clones was done using Freebayes with the options "-p" to set the appropriate ploidy, "-@" to call variants at the previously identified heterozygous positions and the additional options "-m 30, -q 20 -i -X -u" for setting a minimum depth, quality score and to exclude complex variants. Vcfs were then filtered with bcftools to take only SNP variants. As an additional filtering step, the parental and sample vcf files were intersected using bedtools "-intersect" and variants with shared positions were retained from the vcf of the RTG-evolved samples. Next, we used an in-house R script to compare the frequencies of the alternative and reference alleles of each marker in the evolved samples and in the parental ones. We identified putative recombined markers like the ones showing an AF shift comparing the parental and the RTG AF, independently of the direction of the AF shift. Heterozygous markers present in the parental strain but not

genotyped as heterozygous in the RTG samples were considered to have reached a homozygous reference genotype if the position was sequenced. Moreover, the allele of the marker tagged as having an AF shift was compared with the known allele of the marker present in the ancestral strain, and those not matching the expected allele were filtered out. Given the complexity of polyploids analysis, we decided to use a stringent threshold and we annotated regions of AF shift only if they contained at least nine consecutive markers with AF shift regardless of the directionality of the allele that is gained or lost due to recombination. In this way, we avoided calling false-positive AF shift regions at the cost of decreasing our resolution. This approach may decrease the number of gene-conversion events detected as they are generally small (< 2 kbp) and thus may fall below our threshold. Moreover, our resolution in mapping long recombination events was limited by the local heterozygosity between the haplotypes as we did not have information on the phased haplotypes for our strains. Regions of local homozygosity between the two recombining haplotypes may be intervaling regions of heterozygosity on which we identified an AF shift thus breaking the AF shift region and resulting in the fragmented pattern observed. For this reason, we refrained to give numbers of events but only referring to AF shift regions. Among those LOHs/AF shift regions, the ones overlapping for >80% of their length in at least ≥ 70% of the samples considering each datasets separately (wild-type RTG, LYS2/URA3 selected RTG, and ndt80Δ RTG) were filtered out because were considered to be unlikely generated by RTG, but rather representing pre-meiotic events. Additionally, LOH/AF shift regions overlapping for ≥ 90% of their length with the ones found in the T0 samples were also excluded. LOHs spanning the LYS2/URA3 locus on chromosome II were not excluded but that chromosome was not counted for calculating the percentage of markers lying in regions with AF shift. Moreover, LOHs/AF shift regions or CNV already present in the parental samples engineered with the LYS2/URA3 system, or in which NDT80 was deleted, was removed from the respective derived RTG samples. LOHs/AF shift regions in subtelomeric regions were excluded due to the unreliability of mapping in these highly repetitive regions. Finally, the list of annotated CNVs in core parts of the genome of the parental strains was used to filter the LOH/AF shift regions detected. The plots of the AF shift distribution were done using an in-house R script implemented with ggplot2 (v3.6.1) that takes into account the genotype shift of each marker in the recombination blocks.

**Inferring the parental haplotypes from the mother-daughter RTG pairs.** One M-D RTG pair derived from OS1364 was used as a proof of concept to phase a region on the left arm of chromosome IX where recombination likely resulted from a cross-over between two homologs. The genotype of each heterozygous marker was inferred based on the AF shift in the M-D pair. For example, when we detected an AF shift toward the alternative allele in a marker, that is, the RTG sample has gained one alternative allele, and the genotype of the allele was assigned as the reference for that marker. We followed the same approach when we detected an AF shift toward the reference allele and assigned in this case the alternative genotype. Then, the information of these recombined haplotypes was used to infer the third haplotype based on the allele dosage, which is the number of "Ref" or "Alt" copies estimated for each heterozygous marker. For instance, if the initial genotype was "Ref/Alt/Alt" and the AF shift detected was toward the reference allele, the third unknown haplotype harbored an "Alt" allele and the two recombining haplotypes had respectively a "Reference" and "Alternative" allele. The information of the recombined haplotypes was validated on a second RTG M-D pair having a recombination event spanning the same region.

**Microplate cultivations.** The osmotic stress and ethanol tolerance were assessed with microcultures in media containing 25% (w/v) sorbitol and 8% (v/v) ethanol, respectively. The microcultures were carried out in 100-well honeycomb microtiter plates at 25 °C (with continuous shaking), and their growth dynamics were monitored with a Bioscreen C MBR incubator and plate reader (Oy Growth Curves Ab, Finland). The wells of the microtiter plates were filled with 300 μL of YPM medium (1% yeast extract, 2% peptone, 1% maltose) supplemented with sorbitol (25%) and ethanol (8% v/v). Control cultivations in media without sorbitol or ethanol were also carried out. Pre-cultures of the strains were started in 20 mL YPM medium and incubated at 25 °C with shaking at 120 rpm overnight. We measured the optical density at 600 nm, and pre-cultures were diluted to a final $OD_{600}$ value of 3. The microcultures were started by inoculating the microtiter plates with 10 μL of cell suspension per well (for an initial $OD_{600}$ value of 0.1) and placing the plates in the Bioscreen C MBR. The optical density of the microcultures at 600 nm was automatically read every 30 min. Four replicates were performed for each strain in each medium. Growth curves for the microcultures were modeled based on the $OD_{600}$ values over time using the 'GrowthCurver'-package for R[48].

**Flask-scale very high-gravity wort fermentations.** 50 mL-scale fermentations were carried out in 100 mL Schott bottles capped with glycerol-filled airlocks. Yeast strains were grown overnight in 25 mL YPM medium at 25 °C. The pre-cultured yeast was then inoculated into 50 mL of 32 °P wort made from malt extract (Senson Oy, Finland) at a rate of 7.5 g fresh yeast $L^{-1}$. Fermentations were carried out in duplicate at 20 °C for 13 days. Fermentations were manually monitored by weighing mass lost as $CO_2$. The alcohol content of the final beer was measured with

an Anton Paar density meter DMA 5000 M with Alcolyzer beer ME and pH ME modules (Anton Paar GmbH, Austria).

**2L scale high-gravity wort fermentations.** Strains were characterized in fermentations performed in a 20 °P high-gravity wort at 20 °C. Cultures were propagated essentially with the use of a "generation 0" fermentation prior to the actual experimental fermentations[49]. Briefly, yeasts were first grown from frozen stocks 30% glycerol stocks in 100 mL YP-Maltose (1% yeast extract, 2% peptone, 4% maltose) and grown overnight. Then cells were transferred to 500 mL YP-Maltose and incubated overnight at 25 °C with shaking at 120 rpm. Finally, the culture was inoculated into two liters of 15°P wort and grown for 1 week (the 'G0' fermentation) at 20 C. This yeast was then collected and inoculated into the experimental fermentation as described below. The experimental fermentations were carried out in triplicate, in 3-liter cylindroconical stainless steel fermenting vessels, containing 2 liters of wort medium. The 20 °P wort (98 g of maltose, 34.7 g of maltotriose, 24 g of glucose, and 6.1 g of fructose per liter) was produced at the VTT Pilot Brewery from barley malt and malt extract (Senson Oy, Finland). Fermentations were inoculated at a rate of 5 g fresh yeast $L^{-1}$ (corresponding to ~15 × 10⁶ viable cells $mL^{-1}$). The wort was oxygenated to 10 mg $L^{-1}$ prior to pitching (oxygen indicator model 26073 and sensor 21158; Orbisphere Laboratories, Switzerland). The fermentations were carried out at 20 °C until the alcohol level stabilized, or for a maximum of 15 days. Wort samples were drawn regularly from the fermentation vessels aseptically and placed directly on ice, after which the yeast was separated from the fermenting wort by centrifugation (9000 × g, 10 min, 1 °C). Samples for yeast-derived flavor compound analysis were drawn from the beer when fermentations were ended. Cell viability was measured by propidium iodide staining of the cells that were collected at the end of the fermentations using a Nucleocounter ® YC-100™ (ChemoMetec, Denmark).

**Chemical analysis of fermentable sugars.** The concentrations of fermentable sugars (maltose and maltotriose) were measured by HPLC using a Waters 2695 separation module and Waters system interphase module liquid chromatography coupled with a Waters 2414 differential refractometer (Waters Co., Milford, MA, USA). A Rezex RFQ-Fast Acid H⁺ (8%) LC column (100 × 7.8 mm; Phenomenex, USA) was equilibrated with 5 mM $H_2SO_4$ (Titrisol, Merck, Germany) in water at 80 °C, and samples were eluted with 5 mM $H_2SO_4$ in water at a 0.8 mL $min^{-1}$ flow rate. The alcohol level (% vol/vol) of samples was determined from the centrifuged and degassed fermentation samples using an Anton Paar density meter DMA 5000 M with Alcolyzer beer ME and pH ME modules (Anton Paar GmbH, Austria).

**Chemical analysis of aroma compounds in fermented beers.** Yeast-derived higher alcohols and esters were determined by headspace gas chromatography with flame ionization detector (HS-GC-FID) analysis. Four-milliliter samples were filtered (0.45 μm pore size) and incubated at 60 °C for 30 min, and then 1 mL of gas-phase was injected (split mode, 225 °C, split flow of 30 mL $min^{-1}$) into a gas chromatograph equipped with an FID detector and headspace autosampler (Agilent 7890 series; Palo Alto, CA, USA). Analytes were separated on an HP-5 capillary column (50 m by 320 μm by 1.05 μm column; Agilent, USA). The carrier gas was helium (constant flow of 1.4 mL $min^{-1}$). The temperature program was 50 °C for 3 min, 10 °C $min^{-1}$ to 100 °C, 5 °C $min^{-1}$ to 140 °C, 15 °C $min^{-1}$ to 260 °C, and then isothermal for 1 min. Compounds were identified by comparison with authentic standards and were quantified using standard curves. 1-Butanol was used as an internal standard (50 μL of a 5 g/L aqueous solution) and was added to each sample.

**Reporting summary.** Further information on research design is available in the Nature Research Reporting Summary linked to this article.

## Data availability
The phenotype data are available within the Supplementary Data 6, 9, 10, and 11. The short reads sequences generated in this study are available at the SRA of NCBI under the accession code PRJNA770168. Source data are provided with this paper.

## Code availability
Custom R and bash scripts associated to this work can be found at GitHub [https://github.com/SimoneMozzachiodi/UnlockingFunctionalPotentialofPolyploidYeasts].

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

## Acknowledgements

S.M. and G.L.: Agence Nationale de la Recherche (ANR-13-BSV6-0006-01, ANR-18-CE12-0004, ANR-15-IDEX-01, ANR-20-CE12-0020), Fondation pour la Recherche Médicale (EQU202003010413), UCA AAP Start-up Deep tech, CEFIPRA, Association pour la Recherche sur le Cancer (ARCPJA32020070002320). G.L., S.M., and A.N.: convention CIFRE 2016/0582 between Meiogenix and ANRT. K.K. and B.G.: no relevant funding. We thank D'Angiolo M. and Adekunle D. for their critical reading of the manuscript.

## Author contributions

S.M., A.N., G.L., conceived the project, S.M., K.K., A.N., B.G., G.L. designed the experiments, S.M., K.K. performed the experiments and analyzed the data, S.M. and G.L. wrote the paper with input from K.K., B.G., and A.N.

## Competing interests

A.N. and G.L. have a patent application on "Yeast strains improvement method" using return-to-growth (US20150307868A1). S.M. was partially funded by Meiogenix. A.N. is the CSO of Meiogenix. The remaining authors declare no competing interests.
