## [Peer Review File · Nature Communications]

Unlocking the functional potential of polyploid yeastsReviewers' Comments:

Reviewer #1:

Remarks to the Author:

Mozzaciodi and coworkers show here that by using a technique called "return to growth", where yeast cells are induced to initiate meiosis but returned to the mitotic cell cycle by the addition of nutrients, polyploid yeast strains of the type often found in commercial use can be induced to undergo recombination and expose cryptic genetic variation, even though they are unable to produce viable spores by virtue of genetic imbalance in their genomes. While the paper does not provide much in the way of novel biological insight, it does provide an illustration of how a technique that has long been used to study meiotic recombination in mutants unable to complete meiosis can be used for the commercial application of generating phenotypic variation in industrial yeast strains, which should allow improvement without having to resort to the commercially unpopular use of genetic engineering. I leave it to the editors to decide whether Nature Communications is an appropriate journal for a paper which does not really advance scientific understanding but which does provide a potentially useful industrial application.

From a scientific point of view, the experimental work is well-reported and well-executed. The following minor comments are offered by way of improving the manuscript:

1. Lines 93-95, large region of LOH near the rDNA in both strains. Actually, in OS1431, the LOH starts a considerable distance to the right of the rDNA, making it unlikely that genetic instability of the rDNA is responsible for this LOH. Perhaps the sentence in question could be re-written for clarity.
2. Lines 102-103. What is meant by "horizontal gene transfer" here? Usually HGT refers to introduction of genetic material by some means other than breeding. What is the evidence that this region did not come in via crosses? I think that it would be more accurate to describe this region as being "homologous to XXX".
3. Figure 1e. It looks as if the ratio of Hom/Het variants is similar for essential and non-essential genes. Is that the case or not? If it is the case, then perhaps the criteria used to identify "high-impact" variants is not selective for deleterious alleles.
4. Lines 205-6. Suggest "mother-daughter pairs that did not complete bud separation at the time of plating" as more accurate.
5. Line 223 and following. One of the surprising findings, which is not commented on further, is that most of the RTG-induced allele-frequency shifts are reciprocal, which is not what one would expect for gene conversion. However, they also are punctate or occur over relatively short regions, whereas a reciprocal allele frequency shift caused by a crossover should be continuous from the crossover point to the telomere. What do authors think is happening?
6. Line 340. While the claim is that RTG induces random gene shuffling, the paper does not demonstrate that allele frequency changes are randomly distributed in the genome (excepting, of course, the region around the locus used to score RTG). Non-randomness, or regions that rarely undergo AF shifts, might limit the industrial utility of the RTG approach.

Reviewer #2:

Remarks to the Author:

This study builds upon previous work by some of the authors (Mozzaciodi et al., Nature Communications 2021; Laureau, PLoS Genetics 2016) that showed how aborted meiosis can lead to recombination events within a *S. cerevisiae* genome, even in strains that cannot complete a full meiotic cycle (eg because of aneuploidy or sterility due to specific gene dysfunction). In this new study, this RTG technology is expanded to (near) sterile aneuploid /polyploid industrial yeasts. The results show that RTG allows inducing recombination events in these strains, with the resulting variants showing phenotypic variation in fermentation performance and aroma production that may be interesting for the generation of superior industrial yeasts.

Overall, this study is scientifically sound and interesting, and the manuscript is well written. I only have minor comments and questions.

1. It would be a good idea to better acknowledge and discuss previous research in this area. Previous studies by the same senior authors already showed changes in grape most fermentation, consumption of different carbon sources and resistance to various stress factors...(see Mozzachiodi et al., Nat Communications 2021 (supplemental dataset 5 of this paper is of particular interest since it shows very similar principles); Laureau PLoS Genetics 2016). Moreover, Honigberg et al, PNAS 2014 deserves recognition as well. I think it would be useful to discuss the new results in the light of this previous work.
2. Although I do see the potential of the RTG technique for sterile strains, one major limitation compared to a normal sexual cycle is that one is limited to one genome only, which makes it impossible to combine properties from different genetic backgrounds. I think this is an important limitation that should be discussed.
3. The pre-selection of candidate variants based on variation in colony morphology merits critical discussion. Does this not hold the risk that one will always pick up variants that are hit in one of the key genes/pathways influencing colony morphology (in addition to possible other variations)? This may bring undesirable tradeoffs, and might even limit the range of variants that does get picked up. Is this step really necessary? Can the downsides be discussed in a bit more detail? Is it possible to trace back the mutations that are responsible for the shift in colony morphology?
4. The sectoring of colonies is intriguing and perhaps also slightly worrying as it might signal instability. It is unclear to me whether the sectors always originate from the first original mother cell/daughter cell (ie mitosis right after RTG). If so, what is happening here? Do both resulting cells undergo mutations/rearrangements? And, do sectors sometimes also appear later during colony growth (indication instability - see also next comment)? It would be useful to revise the text a bit here to make it more clear and better compare the genetics of the two cell types in sectors within the same colony.
5. Related to the comments above regarding the sectoring - When complex polyploid/aneuploid cells do undergo a sexual cycle, the resulting hybrids are often genetically unstable, with karyotypes changing over many generations after the meiosis. One would expect that this is also the case with RTG. How stable are the variants? Can they be stabilized using directed evolution in selective conditions, as is often done for hybrids?
6. Figure 1: panels a and b are very small and difficult to interpret at their current size
7. Line 230 - "two additional cues" rather than "two additional proofs"?

kevin verstrepen

Reviewer #3:

Remarks to the Author:

In recent years, new methods have been developed to generate yeast diversity. Some of these methods have been just applied to *Saccharomyces cerevisiae*, SCRaMble (PMID: 22572789), to intraspecies *S. cerevisiae* hybrids (admixed/mosaic) using CRISPR/Cas9 (PMID: 34643787), or to interspecies hybrids using iHyPr (PMID: 32350251) or an analogous method (PMID: 34518218). Although, most of them generate Genetically Modified Organisms (GMOs), which GMO strains are difficult to apply to the food sector, iHyPr can be considered non-GMO in particular countries (Ishii and Ishii 2021 Trends in Biotechnology). In any case, these methods, not discussed or commented in the manuscript, seek to find similar results/answers to the authors' questions, which also are biologically and industrially important: i) unlock the functional potential of polyploid yeasts (PMID: 34643787, 32350251, 34518218); ii) improving industrial strains (PMID: 34643787, 32350251, 34518218); iii) or facilitate genotype-phenotype associations, which some of the aforementioned methods has been

proved to be successful (PMID: 34518218).

Here, Mozzachiodi et al. has gone a step further in the utilization of the particular yeast mechanism, return to growth (RTG). During RTG, a cell yeast not committed to meiosis suffers loss of heterozygosity due to recombination between chromosomes, which are fully resolve in mother and daughter cells, keeping the initial genome content. Liti's group has been testing this mechanism, in diploids of *S. cerevisiae*, to generate yeast diversity and facilitate genotype-phenotype associations. In this manuscript, authors successfully tested this mechanism in polyploid (3n and 4n) admixture/mosaic strains (intraspecies hybrids or strains with genome content from different ancestries) of *S. cerevisiae*. They develop a pipeline to use RTG as a non-GMO mechanism to generate diversity and improve industrial strains. I fully agree about the potential of RTG to generate diversity in diploids and now in polyploids, which is well proven in this manuscript, and its utilization in genotype-phenotype studies, which was demonstrated in a previous study, recently accepted for publication in this same journal (reference 17 of this manuscript). What I am less convinced is about authors' success to improve the industrial strains, and about some of the statements used in the main text. Please, see some of my comments below.

Major comments:

- #. Line 77. I do not consider these RTG strains can be categorized as evolved strains. There is not adaptive evolution experiments performed in this study, just a dissection facilitated by genetic manipulation (generating GMOs) or phenotypic differentiation (generating non GMOs).
- #. Line 290: the rationale about strain selection was not clear to me. I do not see the selected strains were the top best, or significantly the best. Can the authors expand their reasoning a bit more? Or apply some statistics to support their statement?
- #. Line 293 "...RTGs performed at least equally well...": what do authors mean with performing? Supplementary Figure 9a: RTG1 produces less alcohol than RTG2 and OS1364 in a specific period of time, requiring more time to produce the same alcohol than the parent and RTG2. This result might be translated in more expenses to the industry. In supplementary Figure 9b: previous conclusion could be right for both RTGs but they are much worse than WLP001. Based on Figure 4 or Supplementary Figure 9, will the authors suggest for commercialization the studied RTGs strains to replace WLP001 or the parents? I think the main conclusion of this paper is the production of diversity using RTG in polyploids, but I do not see an improvement (line 75-76) in global terms. I suggest authors must soften the expectation based on the results shown in the manuscript.
- #. Figure 4: Why do the authors get different values for a dotted black line (panel b), which is expected to be identical genotypes (mother or daughter), no sectoring? It might suggest that there is sectoring with no morphological differences. This must be discussed in the text.

Minor comments:

- #. Please cite other methods. Lines 40-44 look to be a perfect place.
- #. iHyPr and Naseeb's methods can fit in lines 44-46. In both cases, recombinations have been observed between parental genomes.
- #. Could the authors discuss why are there two bands in supplementary Figure 2b gel pictures?
- #. Line 171-174: do the authors have any clue about how LOH and CNVs induced by CRISPR/Cas9 before RTG can influence in next steps of the pipeline? This is not covered and might be an issue.
- #. Supplementary Table 5: define codes in tables. What does D1 mean? And sectoring 1?
- #. Lines 265-267: residual sugars are also high in both OSs strains, which might be indicated in these lines. An increase in residual sugars is not a desirable trait, which goes against their main conclusion of strain improvement through RTG.
- #. Supplementary Table 8: can authors link sample (seq id) with strain id?
- #. Line 306 and Supplementary Figure 4f: what do the authors mean with diversified? I see diversified values in OS1431 as well, but the authors just indicate OS1364. Regarding Figure 4f, can the authors define open and filled colored dots?
- #. Line 429: the DAPI protocol needs more details or cite a reference, to replicate the experiments.
- #. Line 434: give more details about the fluorescence channel and microscope.
- #. It is not clear to me, how do the authors know just one copy of URA3 was removed? I understand

that the most plausible scenario is for the removal of one copy, using the homologous recombination method, but without additional proof, I will put in "quarantine" this conclusion.

#. Without synchronizing the cells, how do the authors know all cells completed the budding and no additional cells started a new budding cycle?

#. In-house scripts might be deposited in a repository, such as figshare or github. They might be relevant to other researchers.

Suggestions

#. Please, could the authors indicate a synonym name (International collection culture) for OS1364 and OS1431?

#. Please, could the authors indicate the clade designation of these two strains from Gallone et al. (PMID: 27610566) and/or Gonçalves et al. (PMID: 27720622)?

#. Line 92 "...suggesting that these two strains....": I think, based on previous publications on OS1364 and OS1431, authors confirmed these strains are two admixed strains. The way is written ("suggested") looks this result is new.

#. Can the authors show a similar Supplementary Figure 3 but now for OS1364?

#. Supplementary Figure 3: i) can the authors include the strain names (smXXX, etc) in Supplementary Figure 3a bottom panels and other relevant figures in main text or supplementary figures?; ii) to improve clarity, it might help to include all copies in the right side of panels b and c; iii) blue and red dots and lines are not defined in the footnote.

#. Line 303: I am not an expert in sensory analysis, but levels above a threshold are detectable, right? So producing 10.5 mg/L of acetaldehyde might be not negligible as authors pointed.

#. Maybe it must be worth to discuss how the authors will make this methodology more highthroughput, at this point it looks it needs too many steps.

#. Please, consider to expand the origin of Kanamycin cassette.

#. Line 592: how was monitored the CO2 loss? Manually (i.e. weighting) or digitally? It needs clarification.

#. Line 630: Was the internal standard added to each sample? It needs clarification.

Minor corrections

#. Line 297: do the authors mean "remove"?

#. Some text indicated ug and other µg.

#. Line 462: What does it mean "for single"?

#. Line 477 add ")" to close the parenthesis.

#. Typo in line 510 "of were" > "were"

#. Section "LOH detection in RTG samples" needs proofreading.

Response letter

We would like to thank reviewers for their constructive feedback. We have changed the manuscript to accommodate the reviewers' viewpoints. A point-by-point response to the reviewers' comments ("replies" [R]) and a detailed description of the corresponding changes made to the manuscript ("actions" [A]) are reported below.

REVIEWER COMMENTS

Reviewer #1 (Remarks to the Author):

Mozzaciodi and coworkers show here that by using a technique called "return to growth", where yeast cells are induced to initiate meiosis but returned to the mitotic cell cycle by the addition of nutrients, polyploid yeast strains of the type often found in commercial use can be induced to undergo recombination and expose cryptic genetic variation, even though they are unable to produce viable spores by virtue of genetic imbalance in their genomes. While the paper does not provide much in the way of novel biological insight, it does provide an illustration of how a technique that has long been used to study meiotic recombination in mutants unable to complete meiosis can be used for the commercial application of generating phenotypic variation in industrial yeast strains, which should allow improvement without having to resort to the commercially unpopular use of genetic engineering. I leave it to the editors to decide whether Nature Communications is an appropriate journal for a paper which does not really advance scientific understanding but which does provide a potentially useful industrial application.

From a scientific point of view, the experimental work is well-reported and well-executed. The following minor comments are offered by way of improving the manuscript:

1. Lines 93-95, large region of LOH near the rDNA in both strains. Actually, in OS1431, the LOH starts a considerable distance to the right of the rDNA, making it unlikely that genetic instability of the rDNA is responsible for this LOH. Perhaps the sentence in question could be re-written for clarity.

[R & A] We thank the reviewer for pointing this out. We have rewritten the sentence to improve its clarity (**Line 100**). We have re-analyzed the heterozygous markers flanking the rDNA in OS1431 and plotted their allele frequencies in the figure below. Indeed, as suggested by the re reviewer, the heterozygous marker configuration is complex with a stretch of roughly 146 kbp close to the rDNA (red line) in which the allele frequency of the alternative allele is equal to 0.5, followed by a stretch of 65 kbp in which the alternative allele has an allele frequency equal to 0.75 or 0.25. Although is difficult to retrace the complex haplotypes history, which is shaped by both the admixture event that originated the tetraploid and the subsequent LOH, it does support a recombination event at the rDNA site.

2. Lines 102-103. What is meant by “horizontal gene transfer” here? Usually HGT refers to introduction of genetic material by some means other than breeding. What is the evidence that this region did not come in via crosses? I think that it would be more accurate to describe this region as being “homologous to XXX”.

[R] We thank this reviewer for raising this point and apologise for not being clear. This specific region together with several other events have been shown to derive from horizontal-gene-transfers (HGT) from non-*Saccharomyces* yeasts that coexist in the fermenting environment [10.1073/pnas.0904673106], [https://doi.org/10.1038/s41586-018-0030-5]. In brief, lines of evidence supporting HGT include that these species do not form viable hybrids with *S. cerevisiae* and the HGT do not replace the homologous region but instead are found close to subtelomeric regions [https://doi.org/10.1038/s41586-018-0030-5]. The specific region detected in OS1364 derived from *Z. bailii* has been identified as region B in other strains [https://doi.org/10.1038/s41586-018-0030-5]. Although we cannot confidently reconstruct the series of events that lead to the extant OS1364 genome, the most likely scenario is that the HGT was present in one of the founder strains that generated this polyploid. Consistently, OS1364 has a partial Wine/European ancestry (https://doi.org/10.1038/s41586-018-0030-5, where the HGT region B has been frequently detected.

[A] We modified the sentence at **line 110** to specify the origin of this HGT region.

3. Figure 1e. It looks as if the ratio of Hom/Het variants is similar for essential and non-essential genes. Is that the case or not? If it is the case, then perhaps the criteria used to identify “high-impact” variants is not selective for deleterious alleles.

[R & A] We used the standard approach provided by Ensembl variant effect prediction to assign a predicted effect based on the variants identified. We followed the reviewer suggestion and calculated the respective ratios which are reported below:

OS1364, Hom/Het (essential): 0.111 Hom/Het (non-essential): 0.287

OS1431, Hom/Het (essential): 0.263 Hom/Het (non-essential): 0.246

The ratio of homo/het variants in OS1364 is more than two-fold higher for non-essential genes whereas OS1431 has a similar ratio between the two. We agree that the abundance of homozygous LOF variants in OS1431 strain is unexpected and we took a closer look. We found that 3/5 are stop-loss affecting dubious open reading frames overlapping another gene in the opposite sense. We report one example below:

Both genes are reported as essential in Liu et al. 2015 and we observed a LOF in *YOL134C*. However, *YOL134C* is a dubious ORF unlikely to encode a functional protein (<https://www.yeastgenome.org/locus/S000005494>) and its essentiality very likely derive from the overlap with *HRT1*. If then we count these 3 overlapping dubious genes as “non essential”, the Hom/Het (essential) ratio in OS1431 goes from 0.263 to 0.105. We have added these informations in **lines 120-121** and in the **Materials & Methods** at **lines 436-439**.

4. Lines 205-6. Suggest “mother-daughter pairs that did not complete bud separation at the time of plating” as more accurate.

[R & A] We implemented this suggestion.

5. Line 223 and following. One of the surprising findings, which is not commented on further, is that most of the RTG-induced allele-frequency shifts are reciprocal, which is not what one would expect for gene conversion. However, they also are punctate or occur over relatively short regions, whereas a reciprocal allele frequency shift caused by a crossover should be continuous from the crossover point to the telomere. What do authors think is happening?

[R] We thank the reviewer for raising this important point. The regions that show reciprocal recombination events but do not extend to the entire chromosome arms are likely crossovers that we could not fully map because of the partial local homozygosity between the recombining haplotypes which breaks the identification of regions with allele frequency shift. As example, the zoom-ins on **Figure 3e** are composed of several reciprocal recombination events, which are separated by regions where the two recombining haplotypes likely share the same genotype where we could not map allele frequency shift regions encompassing at least 9 heterozygous markers. For example, the chromosome X-R event is likely a single recombination that extend to the telomere. For this reason, we have refrained to provide

number of recombination events in this work (in contrast to our work on diploids) and instead provided conservative estimates of fraction of genomes with AF shifts.

In addition to the broken recombination events deriving from heterozygous marker distribution, a genuine biological process that generate large recombination that do not extend up to the telomere consists in double-crossover events (see Figure 3 <https://doi.org/10.1371/journal.pgen.1005781>).

Moreover, given the stringent threshold of 9 consecutive markers used for calling AF shift regions (**Materials & Methods**), it is possible that we underestimate small (< 1-2 kbp) gene conversions. However, this stringent threshold is aimed to reduce the number of false positive calls that will escalate if we reduce the number of consecutive markers.

[A] We now clarify these AF patterns in Figure 3e caption and provide a detailed discussion in the **Materials & Methods** section at **lines 603-609**.

6. Line 340. While the claim is that RTG induces random gene shuffling, the paper does not demonstrate that allele frequency changes are randomly distributed in the genome (excepting, of course, the region around the locus used to score RTG). Non-randomness, or regions that rarely undergo AF shifts, might limit the industrial utility of the RTG approach.

[R] We agree with the reviewer that “random” is not the correct word. What we meant here is that RTG recombination does not occur at a specific locus and in a specific direction (loss or gain of a specific allele) except for the LOH occurring at the locus regulating the morphological trait selected. In our previous work [<https://doi.org/10.1038/s41467-021-26883-8>], we demonstrated that RTG generates LOHs that were associated with meiotic hotspots in agreement with the RTG recombination relying on meiotic DSB. However, it is possible that in specific strains, recessive deleterious alleles may prevent LOH formation in specific directions. If so, such effect would be strain specific and not a general property of the RTG mechanism.

[A] We expanded and clarified our discussion at **lines 375-379** and cited the relevant work.

Reviewer #2 (Remarks to the Author):

This study builds upon previous work by some of the authors (Mozzachiodi et al., Nature Communications 2021; Laureau, PloS Genetics 2016) that showed how aborted meiosis can lead to recombination events within a *S. cerevisiae* genome, even in strains that cannot complete a full meiotic cycle (eg because of aneuploidy or sterility due to specific gene dysfunction). In this new study, this RTG technology is expanded to (near) sterile aneuploid /polyploid industrial yeasts. The results show that RTG allows inducing recombination events in these strains, with the resulting variants showing phenotypic variation in fermentation performance and aroma production that may be interesting for the generation of superior industrial yeasts.

Overall, this study is scientifically sound and interesting, and the manuscript is well written. I only have minor comments and questions.

1. It would be a good idea to better acknowledge and discuss previous research in this area. Previous studies by the same senior authors already showed changes in grape most fermentation, consumption of different carbon sources and resistance to various stress

factors...(see Mozzachiodi et al., Nat Communications 2021 (supplemental dataset 5 of this paper is of particular interest since it shows very similar principles); Laureau PLoS Genetics 2016). Moreover, Honigberg et al, PNAS 2014 deserves recognition as well. I think it would be useful to discuss the new results in the light of this previous work.

[R & A] We thank the reviewer for the suggestion. Accordingly, we added further discussion of previous works at **lines 370-371, lines 375-379, lines 385-387** and we now refer to Honigberg et al., PNAS 1994 in our introduction at **line 67**.

2. Although I do see the potential of the RTG technique for sterile strains, one major limitation compared to a normal sexual cycle is that one is limited to one genome only, which makes it impossible to combine properties from different genetic backgrounds. I think this is an important limitation that should be discussed.

[R] We agree that RTG is especially suited for sterile industrial strains and does not introduce genetic diversity by breeding. This can only be achieved through sexual reproduction which however is impaired in most industrial strains. However, we have previously shown [<https://doi.org/10.1038/s41467-021-26883-8>] that RTG can induce LOH at the *MAT* locus generating diploid hybrids that can mate. This property may be exploited to generate *de-novo* polyploid hybrids by breeding the RTG “maters” to other genetic backgrounds with opposite mating configurations.

[A] We expanded the related discussion at **lines 415-420**.

3. The pre-selection of candidate variants based on variation in colony morphology merits critical discussion. Does this not hold the risk that one will always pick up variants that are hit in one of the key genes/pathways influencing colony morphology (in addition to possible other variations)? This may bring undesirable tradeoffs, and might even limit the range of variants that does get picked up. Is this step really necessary? Can the downsides be discussed in a bit more detail? Is it possible to trace back the mutations that are responsible for the shift in colony morphology?

[R] We thank the reviewer for raising this important point. We agree that the selection on variation of colony morphology is a critical aspect.

It is true that RTG variants shared a common region of LOH when selected for a specific phenotype (e.g. in **Supplementary Figure 7**). However, this was not the case for the RTGs with a wrinkled phenotype in both OS1364 and OS1431 where we could not map a common region across the M-D RTGs. Moreover, we have never retrieved the same genome-wide recombination pattern despite picking RTG sectors using the same colony phenotypes (**Supplementary Figure 6**). In contrast, the white/red phenotypes in both RTG M-D have recombination at the *ADE1* locus. We observed a non-synonymous substitution in 2 out of 3 alleles in R264K, predicted to be deleterious with Mutfunc <http://www.mutfunc.com/results/yceaea506?f=P27616#pv>. This allele becomes homozygous in the red RTG sectors upon recombination and likely is responsible for the phenotype.

To evaluate more globally the potential trade-offs arising as a result of the colony phenotype selected, we screened the phenotypic variability in the RTG sectors of OS1431 and OS1364 dividing the M-D pairs according to the phenotype selected (OS1431: wrinkled/smooth, OS1364: darker/white). We choose these two classes as both have at least 5 RTG M-D pairs

allowing for a statistical test. We found that the colony phenotype selected introduced a detectable variation only in the growth rate (R) in sorbitol in OS1431 derived sectors, and in the growth rate (R) in maltose for the OS1364 derived sectors (Wilcoxon-test two tailed). However, in both cases the median phenotype of the worst performers group was close to that of the relative parental strain.

The above results suggested that the colony phenotype may be linked to deleterious trade-offs but in our datasets the worsened phenotype was more a strain specific feature rather than a property of a specific selected colony morphology phenotype.

As anticipated by the reviewer, the colony morphology selection is not strictly necessary, but it allows an efficient identification of true RTG colonies. An alternative approach is to restrict the usage of the colony morphology as a proxy to follow the appearance of RTG colonies and perform HT genomic and/or phenotype screening on large cohorts of colonies to identify RTGs that have not change morphology. Such approach is less effective since will require large screenings, but overall feasible given the available modern technologies.

[A] To clarify the potential consequences of the colony morphology screen, we added (1) A discussion on the downside of the colony-morphology selection at **lines 387-391**. (2) A description of the results obtained at **lines 294-299** and a new panel in **Supplementary Figure 8a**. (3) The information on the phenotypic classes used for the plot was added to **Supplementary table 6**.

4. The sectoring of colonies is intriguing and perhaps also slightly worrying as it might signal instability. It is unclear to me whether the sectors always originate from the first original mother cell/daughter cell (ie mitosis right after RTG). If so, what is happening here? Do both resulting cells undergo mutations/rearrangements? And, do sectors sometimes also appear later during colony growth (indication instability - see also next comment)? It would be useful

to revise the text a bit here to make it more clear and better compare the genetics of the two cell types in sectors within the same colony.

[R] We thank the reviewer for pointing to this aspect. We have reported in the main text that sectoring colonies were never without sporulation induction (**lines 221-222**). We proved that the sectors originate from the first mother cell/daughter cell pair according to the following evidence. First, each sector comprises roughly half of the colony as expected if that is derived from the first mitotic division. Second, the two sequenced sectors have largely reciprocal genome-wide recombination landscape typical of RTG cells, these can be generated only upon the first mitotic division after meiotic abortion. Third, the sectors never arise in the periphery of the growing colony, as expected in case of ongoing genome instability during the clonal expansion [e.g. DOI: 10.1126/science.1087706]. Although the sector area is not always exactly half of the colony, the variation produced is compatible with differences in growth rate of the M-D RTG pairs.

Motivated by the reviewer comments, we have now tested the stability of the colony morphology phenotype by streaking some of the RTG variants and observed that all the colonies maintained the selected phenotype.

[A] We made several edits in the relative section of the results “Selection of recombined RTG clones by natural colony phenotypes” to improve its clarity.

5. Related to the comments above regarding the sectoring - When complex polyploid/aneuploid cells do undergo a sexual cycle, the resulting hybrids are often genetically unstable, with karyotypes changing over many generations after the meiosis. One would expect that this is also the case with RTG. How stable are the variants? Can they be stabilized using directed evolution in selective conditions, as is often done for hybrids?

[R & A] We know from previous studies (Laureau et al., 2016, Mozzachiodi et al., 2021) that the RTG process itself is not mutagenic. We have not measured the long-term stability of the RTG variants, but overall RTG clones should reflect similar mutational rates of the parental strains. One possible source of genomic instability could arise if LOHs uncover deleterious recessive genes involved in genome maintenance but in that case the instability would be specific of that RTG variant.

We are not aware of any experiment in which RTG variants have been propagated through directed evolution in selective conditions and indeed could be an interesting future angle for integrating these two different approaches and this aspect is discussed (lines 411-413).

6. Figure 1: panels a and b are very small and difficult to interpret at their current size

[R & A] We modified Figure 1 to increase the size of panels a and b.

7. Line 230 – “two additional cues” rather than “two additional proofs”?

[R & A] We corrected this.

kevin verstrepen

Reviewer #3 (Remarks to the Author):

In recent years, new methods have been developed to generate yeast diversity. Some of these methods have been just applied to *Saccharomyces cerevisiae*, SCRaMbLE (PMID: 22572789), to intraspecies *S. cerevisiae* hybrids (admixed/mosaic) using CRISPR/Cas9 (PMID: 34643787), or to interspecies hybrids using iHyPr (PMID: 32350251) or an analogous method (PMID: 34518218). Although, most of them generate Genetically Modified Organisms (GMOs), which GMO strains are difficult to apply to the food sector, iHyPr can be considered non-GMO in particular countries (Ishii and Ishii 2021 Trends in Biotechnology). In any case, these methods, not discussed or commented in the manuscript, seek to find similar results/answers to the authors' questions, which also are biologically and industrially important: i) unlock the functional potential of polyploid yeasts (PMID: 34643787, 32350251, 34518218); ii) improving industrial strains (PMID: 34643787, 32350251, 34518218); iii) or facilitate genotype-phenotype associations, which some of the aforementioned methods has been proved to be successful (PMID: 34518218).

Here, Mozzachiodi et al. has gone a step further in the utilization of the particular yeast mechanism, return to growth (RTG). During RTG, a cell yeast not committed to meiosis suffers loss of heterozygosity due to recombination between chromosomes, which are fully resolve in mother and daughter cells, keeping the initial genome content. Liti's group has been testing this mechanism, in diploids of *S. cerevisiae*, to generate yeast diversity and facilitate genotype-phenotype associations.

In this manuscript, authors successfully tested this mechanism in polyploid (3n and 4n) admixture/mosaic strains (intraspecies hybrids or strains with genome content from different ancestries) of *S. cerevisiae*. They develop a pipeline to use RTG as a non-GMO mechanism to generate diversity and improve industrial strains. I fully agree about the potential of RTG to generate diversity in diploids and now in polyploids, which is well proven in this manuscript, and its utilization in genotype-phenotype studies, which was demonstrated in a previous study, recently accepted for publication in this same journal (reference 17 of this manuscript). What I am less convinced is about authors' success to improve the industrial strains, and about some of the statements used in the main text. Please, see some of my comments below.

Major comments:

#. Line 77. I do not consider these RTG strains can be categorized as evolved strains. There is not adaptive evolution experiments performed in this study, just a dissection facilitated by genetic manipulation (generating GMOs) or phenotypic differentiation (generating non GMOs).

[R] We respectfully ask to keep the term “evolved” for the following reasoning. Experimental evolution is not restricted to adaptive evolution, for example mutation accumulation lines are considered laboratory evolution experiments even nearly no selection is applied there [doi:10.1038/nrg3564] and these lines are considered evolved when compared to the ancestral founder strain. We agree with the reviewer that our protocol has a very limited number of mitotic generation but we feel the term “evolved” is justified by the genomic and phenotypic variation of the RTG clones compared to their respective ancestral founder strains.

#. Line 290: the rationale about strain selection was not clear to me. I do not see the selected strains were the top best, or significantly the best. Can the authors expand their reasoning a bit more? Or apply some statistics to support their statement?

[R] We apologize that the selection of strains was not made clear. Regarding the two RTGs derived from OS1364, we selected the first and the third top ABV producers given the results obtained in the laboratory scale wort fermentation (**Supplementary Table 9**). The second top best was excluded because it was bearing an aneuploidy on chromosome VIII (**Supplementary Figure 7b, Supplementary Table 9**). We selected the two RTGs derived from OS1431 because they produced the highest ABV at the end of the laboratory scale wort fermentation (**Supplementary Table 9**) and were the M-D RTG pairs with the highest phenotypic variability (**Supplementary Figure 8**).

[A] We expanded in the caption of **Supplementary Figure 8** the description of the selection process of the RTG pairs used for the follow-up fermentation experiment.

#. Line 293 “...RTGs performed at least equally well...”: what do authors mean with performing? Supplementary Figure 9a: RTG1 produces less alcohol than RTG2 and OS1364 in a specific period of time, requiring more time to produce the same alcohol than the parent and RTG2. This result might be translated in more expenses to the industry. In supplementary Figure 9b: previous conclusion could be right for both RTGs but they are much worse than WLP001. Based on Figure 4 or Supplementary Figure 9, will the authors suggest for commercialization the studied RTGs strains to replace WLP001 or the parents? I think the main conclusion of this paper is the production of diversity using RTG in polyploids, but I do not see an improvement (line 75-76) in global terms. I suggest authors must soften the expectation based on the results shown in the manuscript.

[R] We thank the reviewer for pointing this. By “equally well” we mean that at the end of the fermentation process the RTGs and the parental strains have produced similar levels of ABV (The differences between the parent and the two RTG were 0.08% and 0.02% ABV respectively). We did not compare the ABV before the fermentation was completed which took roughly 7 days, a common time for beer fermentation, but we agree that if the fermentation was interrupted at day 4, for instance, RTG1 would have produced less alcohol than the parental strain (at day 4, difference of 0.64% and at day 5 difference of 0.14%). The same reasoning applies to OS1431 and relative RTGs.

We included WLP001 to have a commercial strain used in beer fermentation with which we could compare our parental strains, but we did not intend to replace WLP001 with our strains. Instead, we propose to generate RTG variants that can replace the respective parental strain, in this case OS1364 or OS1431, to which the RTG variants should be compared to evaluate their phenotypic diversification. A key improvement is the increase of post-fermentation viability in 1 of the 4 RTG screened compared to the respective parent. In addition, the variability in the aroma profile of some of the RTG variants can be exploited to produce beer with different aromas.

[A] To improve clarity, we explained what we mean by “equally well” at **lines 316-317** and we edited **lines 316-317** to explain the performance of sm244 compared to the parent during the fermentation process.

#. Figure 4: Why do the authors get different values for a dotted black line (panel b), which is expected to be identical genotypes (mother or daughter), no sectoring? It might suggest that there is sectoring with no morphological differences. This must be discussed in the text.

[R] We thank the reviewer for pointing this out. We apologize for the incomplete sentence in the caption. The dotted lines mark M-D RTG pairs in which only one of the sectors was sequenced.

[A] We fixed the caption of **Figure 4** specifying that those samples were M-D RTG sectors from which only half of the sector was sequenced (the mother or the daughter) but both sectors were phenotyped.

Minor comments:

#. Please cite other methods. Lines 40-44 look to be a perfect place.

#. iHyPr and Naseeb’s methods can fit in lines 44-46. In both cases, recombinations have been observed between parental genomes.

[R & A] We followed the reviewer's suggestions and expanded the discussion of other methods that have been developed to improve industrial strains or designed hybrids at **lines 43-46**.

#. Could the authors discuss why are there two bands in supplementary Figure 2b gel pictures?

[R & A] The higher band found in the first well is generated after electrophoretic run and staining in Ethidium bromide. In contrast the second band in the 3rd well may be due to small insertion arising upon CRISPR/Cas9 deletion of *SPO11* in one of the 3 haplotypes. By using the band in the second well as comparison we can estimate that the size of this event is smaller than 60 bp, so we can still conclude that *SPO11* is not functional in agreement with a lack of induced recombination upon RTG. We added a note specifying that in **Supplementary Figure 2b**.

#. Line 171-174: do the authors have any clue about how LOH and CNVs induced by CRISPR/Cas9 before RTG can influence in next steps of the pipeline? This is not covered and might be an issue.

[R & A] Regarding the bioinformatic approach, markers lying in LOH and CNV regions induced upon CRISPR/Cas9 were removed from the counting of markers having an allele frequency shift (AF), we specify that at **lines 616-618**.

Regarding the experimental approach, the LOH and CNV are not present on chromosome II where they could alter the selection of recombination events and the measuring of *URA3*-loss rates. We used CRISPR/Cas9 mutants only to validate our initial hypothesis on the potential of industrial polyploids to perform RTG but the phenotypic screening was restricted to non-GMO RTG as indicated in lines 285-287.

#. Supplementary Table 5: define codes in tables. What does D1 mean? And sectoring 1?

[R & A] We thank the reviewer for pointing this out. D1 refers to YPD with 1% dextrose (**Materials & Methods**). We added a footnote in **Supplementary table 5**. Sectoring 1 refers to how many colonies with sectors were counted per plate. We now specify that in the header of the column.

#. Lines 265-267: residual sugars are also high in both OSs strains, which might be indicated in these lines. An increase in residual sugars is not a desirable trait, which goes against their main conclusion of strain improvement through RTG.

[R & A] We agree with the reviewer that an increase in residual sugar above a certain threshold is not desirable. Comparing the RTGs to their respective parents, only 1 RTG has increased residual maltose (sm244, t-test p-value=0,01), whereas other 2 have lower residual maltose compared to their parent(sm399, sm408, t-test p-value=0,002 and p-value=0,042). The other residual sugars do not show a significant difference. However, given the quantity of residual sugars compared to the initial quantity in the fermentation (98 g/L Maltose, 34.8 g/L Maltotriose, 24 g/L Glucose, 6.1 g/L Fructose) the differences are not significantly higher. In our case, residual sugars fall in ranges found in commercial beers [<https://doi.org/10.1016/j.foodchem.2013.07.008>] [<https://doi.org/10.1002/jsfa.10337>] or lower compared to fermentations performed in other studies [10.1007/s00253-016-7588-3]. In addition, our fermentation was performed at a high sugar concentration (20°P), which will also impact the level of residual sugars.

We now also report the percentage of residual sugars in **Supplementary Table 11** and the results for all the three replicates.

#. Supplementary Table 8: can authors link sample (seq id) with strain id?

[R & A] We added the respective information to **Supplementary Table 8**.

#. Line 306 and Supplementary Figure 4f: what do the authors mean with diversified? I see diversified values in OS1431 as well, but the authors just indicate OS1364. Regarding Figure 4f, can the authors define open and filled colored dots?

[R & A] We thank the reviewer for pointing it out. By “diversified” we meant that the two RTGs were showing variability in their aroma profile compared to the respective parental strain. We now specify that at **lines 334-336**. The definition of the colored dots is given in the figure legend on the right as follows. Filled: Above Threshold. Empty: Below Threshold. We slightly

changed the legend on the right of **Figure 4f** and specified that, also in the caption of **Figure 4f**.

We expanded the description at **lines 334-336** regarding the sensory profile variation in the fermentations of the RTG derived from OS1431, which was limited only to variability in 2-Phenylethylacetate. We replaced **Figure 4f** with the same plot where the y-axis limits were moved from 0 (min), 2 (max) to 0.5 (min), 1.5 (max) to improve the readability.

#. Line 429: the DAPI protocol needs more details or cite a reference, to replicate the experiments.

#. Line 434: give more details about the fluorescence channel and microscope.

[R & A] We added more information regarding the experimental protocol and the microscope used at **lines 484-488**.

#. It is not clear to me, how do the authors know just one copy of *URA3* was removed? I understand that the most plausible scenario is for the removal of one copy, using the homologous recombination method, but without additional proof, I will put in “quarantine” this conclusion.

[R & A] The scheme depicted in **Figure 2a** shows that the strains were engineered by removing all *URA3* copies and next, one copy of *URA3* was re-introduced at the *LYS2* locus through homologous recombination. We could infer from the sequencing depth of the engineered parental OS1364^{LYS2/URA3} and OS1431^{LYS2/URA3} that they harbour only one *URA3* copy because the local coverage within the boundaries of *URA3* correspond respectively to $\sim 1/3$ and $\sim 1/4$ of the median coverage as shown in the plots below (Left sm291, Right sm292).

The RTG samples obtained from the *URA3*-loss assay were confirmed to be auxotroph for uracil, in addition local coverage of *URA3* goes to 0 in agreement with that the homologous recombination has removed the *URA3* allele upon RTG.

#. Without synchronizing the cells, how do the authors know all cells completed the budding and no additional cells started a new budding cycle?

[R] When we plated our culture after the first budding upon RTG, it is possible that some cells have budded more than once. Our aim was to plate cells that have not completed the first budding cycle in order to retain the M-D RTG cells. When the plating was performed at a later time point, it is possible that some cells have performed a second division but in this case it

would not affect the recognition of M-D RTG cells since this information would have been already lost at the level of the first budding.

#. In-house scripts might be deposited in a repository, such as figshare or github. They might be relevant to other researchers.

[R] We deposited scripts used for the analysis of RTG samples and identification of AF shift regions on a github repository:

<https://github.com/SimoneMozzachiodi/UnlockingFunctionalPotentialofPolyploidYeasts>

Suggestions

#. Please, could the authors indicate a synonym name (International collection culture) for OS1364 and OS1431?

[R & A] We added this information in **Supplementary Table 1** along with the strain id.

#. Please, could the authors indicate the clade designation of these two strains from Gallone et al. (PMID: 27610566) and/or Gonçalves et al. (PMID: 27720622)?

[R & A] We followed the reviewer's suggestion. The strains used here were not sequenced in the indicated published works. Therefore, we used the information from the strains that were phylogenetically close to OS1364 and OS1431 and present in the suggested works, to map our strains in the clades defined. It was not possible to map OS1364 in the clades identified in Gonçalves because the Mosaic Beer clade (Peter et al., 2018)/ Beer 2 clade (Gallone et al., 2016), are not present in that work. OS1364 belongs to the Beer2 clade from Gallone et al., 2016. OS1431 belongs to the Beer1 clade identified in Gallone et al., 2016, and in the English-Irish Ale Clade in the Beer strains analysed in Gonçalves et al., 2016. We added this information to **Supplementary Table 1**.

#. Line 92 "...suggesting that these two strains...": I think, based on previous publications on OS1364 and OS1431, authors confirmed these strains are two admixed strains. The way is written ("suggested") looks this result is new.

[R & A] We agree with the reviewer and have replaced the term "suggesting" with "confirming" so as to not create any confusion in the reader.

#. Can the authors show a similar Supplementary Figure 3 but now for OS1364?

[R & A] We have generated the plots for OS1364 wild-type, OS1364^{LYS2/URA3} and two derived RTGs reporting the sample with the two aneuploidies mentioned in the text at **line 185**.

a

#. Supplementary Figure 3: i) can the authors include the strain names (smXXX, etc) in Supplementary Figure 3a bottom panels and other relevant figures in main text or supplementary figures?; ii) to improve clarity, it might help to include all copies in the right side of panels b and c; iii) blue and red dots and lines are not defined in the footnote.

[R & A] i) We added the strain names to **Supplementary figure 3 a and b**. We added the strains name to the plots in **Supplementary figure 2d, Supplementary figure 5c and d, Supplementary figure 6a and b, Supplementary figure 7b, Figure 4e, Supplementary figure 9b** and to the caption of **Supplementary figure 8c and d**. We did not add where was already indicated in the figure caption e.g. **Figure 3c** or **Supplementary Figure 9a and b**.

ii) We agree with the reviewer's suggestion and added all the copies of chromosome XIV and their relative number changes in panel b and c of **Supplementary figure 3**.

iii) We thank the reviewer for spotting this. Now we have defined that in the caption of **Supplementary figure 3**.

#. Line 303: I am not an expert in sensory analysis, but levels above a threshold are detectable, right? So producing 10.5 mg/L of acetaldehyde might be not negligible as authors pointed.

[R & A] We apologize if this point was not made clear. The sensory thresholds to which we refer here are derived from Meilgaard et al., 1982, the same author points out that sensory thresholds can be highly variable between individuals (Meilgaard et al., 1982, Meilgaard et al., 1993). Moreover, this sensory threshold is not meant to be a strict cut-off for a desirable undesirable flavor as the blend of all the aromatic profiles will produce the sensory profile and this will depend on the style of the beer produced. For instance, WLP001 which is a commonly used commercial strain, produced 21.47 mg/L of Acetaldehyde. However, we do not suggest that the beer produced is not good and we believe that our conclusion holds. Nevertheless we added at **line 329-330** a specification on the variability of the sensory threshold among individuals to clarify this aspect for the non expert readers.

#. Maybe it must be worth to discuss how the authors will make this methodology more highthroughput, at this point it looks it needs too many steps.

[R] We thank the reviewer for the interest in this aspect. The protocol relies on performing a meiotic progression study of the given strain for selecting a specific time point or a time-frame in which to induce RTG. Once that is determined the sporulation culture can be plated at the selected time-point across multiple conditions in a single experiment and screened in a high-throughput way using an approach similar to the one we cited in our discussion (Ruusuvaori, P. *et al.* Quantitative analysis of colony morphology in yeast. *Biotechniques* 56, (2014)) which uses a cell sorter to array cells on a plate, or high-throughput approaches such as single-cell plating module of Singer rotor.

#. Please, consider to expand the origin of Kanamycin cassette.

[R & A] We thank the reviewer for his suggestion. We added this information at **lines 498-499**.

#. Line 592: how was monitored the CO₂ loss? Manually (i.e. weighting) or digitally? It needs clarification.

[R & A] The CO₂ was monitored manually, we added that information in the **material & methods** section at **line 658-659**.

#. Line 630: Was the internal standard added to each sample? It needs clarification.

[R & A] Yes, 1-butanol was added to each sample and standard (50 μ L of a 5 g/L aqueous solution). We added this information at **lines 697-698**.

Minor corrections

#. Line 297: do the authors mean “remove”?

[R & A] We changed that.

#. Some text indicated ug and other μ g.

[R & A] We corrected the two relevant typos.

#. Line 462: What does it mean “for single”?

[R & A] Streak for isolating single colonies. Now we modified it and just referred to it as a streak removing the jargon.

#. Line 477 add “)” to close the parenthesis.

[R & A] We added that.

#. Typo in line 510 “of were” > “were”

[R & A] We corrected that.

#. Section “LOH detection in RTG samples” needs proofreading.

[R & A] We followed the reviewer's suggestion and edited this section to improve its clarity.

Reviewers' Comments:

Reviewer #1:

Remarks to the Author:

The authors have done a good job of addressing my concerns, in particular by adding additional explanatory text. Like reviewer 3, I do not particularly like the use of the term "evolved", but perhaps authors could consider adding a short sentence or phrase explaining the specific meaning they intend. Other than this, I see no reason not to go forward with the manuscript.

Reviewer #2:

Remarks to the Author:

The authors have answered all my comments and edited the manuscript accordingly. I do not have further comments and congratulate the team on a nice paper.

Reviewer #3:

Remarks to the Author:

I highly appreciate that the authors have addressed all my comments. Just one typo was detected in the main text:

#. Line 44: italics for "Saccharomyces".

We thank the reviewers for their previous feedback and are delighted of their positive evaluation of the revised manuscript. The two suggested edits are detailed below.

REVIEWER COMMENTS

Reviewer #1 (Remarks to the Author):

The authors have done a good job of addressing my concerns, in particular by adding additional explanatory text. Like reviewer 3, I do not particularly like the use of the term "evolved", but perhaps authors could consider adding a short sentence or phrase explaining the specific meaning they intend. Other than this, I see no reason not to go forward with the manuscript.

We added a short explanatory sentence at lines 63-66.

Reviewer #2 (Remarks to the Author):

The authors have answered all my comments and edited the manuscript accordingly. I do not have further comments and congratulate the team on a nice paper.

Reviewer #3 (Remarks to the Author):

I highly appreciate that the authors have addressed all my comments. Just one typo was detected in the main text:

#. Line 44: italics for "Saccharomyces".

We corrected the missing italics.